# Solar Irradiance Reduction Using Optimized Green Infrastructure in Arid Hot Regions: A Case Study in El-Nozha District, Cairo, Egypt

**Wesam M. Elbardisy** [1,*] **, Mohamed A. Salheen** [1] **and Mohammed Fahmy** [2]

1   Faculty of Engineering, Ain Shams University, Abbaseya, Cairo 11517, Egypt;
    mohamed_salheen@eng.asu.edu.eg
2   Department of Architecture Engineering, Military Technical Collage, Kobry El-kobba, Cairo 11662, Egypt;
    md.fahmy@mtc.edu.eg
*   Correspondence: welbardisy@eng.asu.edu.eg; Tel.: +20-1001234746

**Abstract:** In the Middle East and North Africa (MENA) region, studies focused on the relationship between urban planning practice and climatology are still lacking, despite the fact that the latter has nearly three decades of literature in the region and the former has much more. However, such an unfounded relationship that would consider urban sustainability measures is a serious challenge, especially considering the effects of climate change. The Greater Cairo Region (GCR) has recently witnessed numerous serious urban vehicular network re-development, leaving the city less green and in need of strategically re-thinking the plan regarding, and the role of, green infrastructure. Therefore, this study focuses on approaches to the optimization of the urban green infrastructure, in order to reduce solar irradiance in the city and, thus, its effects on the urban climatology. This is carried out by studying one of the East Cairo neighborhoods, named El-Nozha district, as a representative case of the most impacted neighborhoods. In an attempt to quantify these effects, using parametric simulation, the Air Temperature ($T_a$), Mean Radiant Temperature ($T_{mrt}$), Relative Humidity (RH), and Physiological Equivalent Temperature (PET) parameters were calculated before and after introducing urban trees, acting as green infrastructure types that mitigate climate change and the Urban Heat Island (UHI) effect. Our results indicate that an optimized percentage, spacing, location, and arrangement of urban tree canopies can reduce the irradiance flux at the ground surface, having positive implications in terms of mitigating the urban heat island effect.

**Keywords:** tree microclimate; thermal comfort; urban heat island; ENVI-met; solar irradiance

## 1. Introduction

### 1.1. MENA Region Climates

Worldwide climate change has become an experienced reality since the advent of industrialization [1,2]. Urbanization and its correlated activities are the root cause of Climate Change (CC), as higher and higher carbon gas emission concentrations continue to be recorded, with a consequently elevated ecological footprint. Campbell-Lendrum and Corvalán discussed the emerging threats of CC, with respect to human health and rapid economic development. Setting policies related to air pollution, the Urban Heat Island (UHI) effect, sanitation, and population density can provide great opportunities to reduce carbon emissions [3]. Huq et al. referred to cities with low-to-middle income— a category most cities in developing countries fall into—as being exposed to the risks related to CC and GHG emissions. The urge to take actions towards reducing emissions, rather than adaptation, are essential, as the impacts are witnessed after a span of 10 to 20 years [1]. Actions at the local level are essential: knowledge, legitimacy, and capacity secure communities against the health-related problems of extreme heat waves [4], foster a sense of responsibility, and allow them to share in decision-making processes [5].

Due to the Middle East and North Africa (MENA) region's hot and arid local climate, the region is expected to suffer from intensive heat waves in an unprecedented manner in the future. It has already been clearly witnessed in metropolis cities, where urbanization and population density are pronounced. Temperatures in both the day and night are expected to increase greatly, with an average temperature ranging from 43 °C to approx. 46 °C by the mid-century, reaching almost 50 °C by the end of the century [4]. The UN Climate summit, held in Paris in 2015, noted that the projected global temperature should be decreased by 2 °C, by following the climate agreements. However, the local climate of the MENA region, according to the Köppen–Geiger Climate Classification [6], will still suffer from accelerated heat increases; in addition, the urban heat island effect factor will contribute to elevating temperatures to higher degrees [7].

Taking the above into account, different urban modeling and projection tools have indicated the urgency of taking actions in these countries. The region will suffer from water scarcity due to dryness, affecting farming and the production of crops, leading to the loss of forests, and facilitating extreme events such as droughts and floods [4]. Hence, not all cities in the MENA region are capable and can survive these impacts that the world has witnessed, is witnessing, and will witness. Yet, more than 50% of the region has dismissed the Green Climate Funds, aimed to develop mitigation and implementation strategies from a total fund of 100 billion [8].

### 1.2. Urban Trees

Trees are prioritized over other vegetative elements, as they attenuate the local temperature, act as passive coolers for the climate character [9], and moderate solar radiation through shading and evapotranspiration. Shading trees can positively reduce ground and surface temperatures and change the energy balance of the surrounding environment [10]. This, in turn, reduces the energy demand for indoor climates and promotes human thermal comfort [11].

The Tree Canopy (TC) intensely shields direct radiation coming from the sun, observed in hot sunny regions such as those in Egypt. The chance of direct and indirect radiation reaching the ground is high when the humidity is low and the sky is clear. In the summer, leaves in the TC, through photosynthesis processes, allow a maximum of 30% of the radiation to fall on the ground or to be transmitted, as most of the radiation is being absorbed. Meanwhile, in the winter, the radiation falling on the ground may differ in the range of 10–80%, depending on the type and character of the tree. This change in the energy budget is highly dependent on the density of the TC leaves, which can enhance the microclimate [12,13]. Additionally, the tree's impact is extended when placed next to buildings at a distance of 2–3 m, as radiation reduction on walls could reach up to 25% with the aid of surrounding materials and surfaces. Furthermore, their impact is clear in parking lots, where the surface temperature of the present materials can be reduced [14].

The evapotranspiration effect moisturizes the space when trees are well-irrigated and appropriately selected [15], and reduces the temperature by up to 5%. Not only trees, but also grass and other vegetative materials can attenuate the temperature by 3%, when compared to barren areas. The evaporation process requires energy from direct solar radiation and the ambient temperature. The vapor pressure difference between the surface and air assists the whole process, and wind continuously replaces saturated air with dry air [16]. Thus, the local microclimate, including direct radiation, wind, humidity, and temperature, highly impacts the evaporation process [12]. In addition, the water available from irrigation or any water resource and the shade provided by the canopy [17] add to the effect.

Transpiration occurs in leaves and plant tissue when they transpire vapor, which is lost to the air. When the plant is watered/irrigated, water and nutrients move through the roots to reach the plant, where vaporization occurs in the leaves. Predominantly, vaporization occurs in the intercellular spaces within the leaf; afterwards, the stomata allow the vapor to pass outside the leaves. Most of the water is lost through the transpiration process, and

a small amount is left within the plant. The transpiration rates vary from one plant species to another, as well as for the plant itself within its growing period. Transpiration is highly affected by the soil characteristics, which influence the amount of water available [18].

Trees also furnish the space, visually and aesthetically, when considering their scale and proportion in the spatial distribution. Contrast is added when distinct colors are present in a space. Trees have other benefits; for example, they replace carbon gas emissions with oxygen, purify the air, shield from noise, and provide security and privacy, especially in private and semi-public spaces [19]. They also increase the land value with respect to the real-estate market [20].

Tree clusters/groupings modify wind patterns, especially in hot arid regions [21]. A well-designed and studied arrangement of trees is required. A certain grouped tree planting type may not be promoted when it acts as a barrier that shields and blocks air flow, compared with open space design. Grouped trees can significantly reduce the Sky View Factor (SVF) during daytime and shield from solar radiation; however, the trapped heat under the tree's canopy slowly releases during the night, promoting higher air temperatures, surface temperatures, and humidity levels in the context [16,21,22]. Figure 1 summarizes the parameters that influence the impact of the trees in urban spaces.

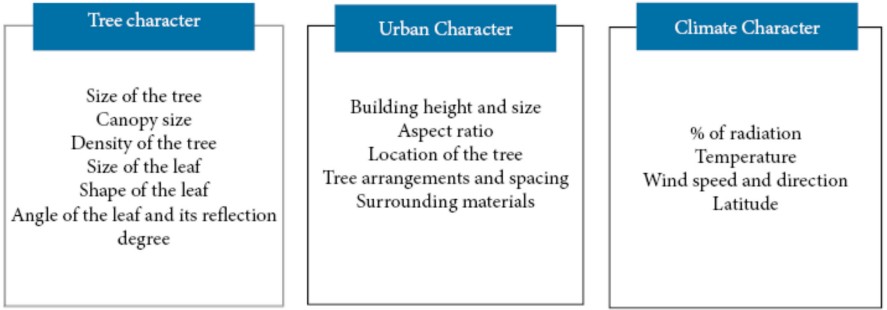

**Figure 1.** Parameters that influence the impact of trees in urban spaces.

*1.3. Mitigation Strategies: Numerical Assessment of Different Proposed Tree Patterns*

Wang and Akbari studied tree types, sizes, and spacings in highly dense residential areas of Montreal, Canada. The sector was characterized by tall buildings, mostly above 15 floors, and the ground surfaces were asphalt. Scenarios were proposed and assessed using the numerical simulation tool ENVI-met; as a result, different tree types, spaces, and sizes were proposed. The results confirmed a general temperature ($T_a$) reduction when implementing a designed tree pattern, which was strongly observed in the midday period. The size of the canopy had a great influence on $T_{mrt}$ and SVF, and the spacing between trees impacted both day- and night-time temperature ranges [23].

Considering a city in Arizona, which has the same climatic zone as in Egypt (hot and arid) [6], Zhao, Sailor, and Wentz aimed to find the optimal tree arrangement to maximize the effect with minimal cost and water consumption. They set different tree arrangements and numerically assessed them through ENVI-met. Trees were arranged in equidistant, clustered, and dispersed arrangements. Their findings confirmed that the benefits of climate moderation in equidistant arrangements outweighed other arrangement types, while the worst was observed for the dispersed arrangement. A high density of trees positively impacts the temperature range and the thermal comfort level; however, overlapped canopies are not recommended, and effective ventilation is required to create wind corridors, which foster radiation exchange. Cool spots have an impact at the whole neighborhood level, and trees next to the buildings are highly recommended. The best placements were in the South-west corner of the building and West direction, with equidistant trees [24].

Furthermore, Amani-Beni et al. have confirmed that clustered trees with grass below have a prime effect on the microclimate, when compared to single trees, grass, and water



separately, in the day- and night-time. Other essential factors should be considered to sustain the effect, such as a regular irrigation pattern, land type, and shape [16]. The utilization of vegetation has been studied by Abu Ali et al., in a low-density residential area of Abu Dhabi, Dubai. The study showed that the cooling pattern is highly influenced by the size of the tree: taller trees perform better in the day-time; however, the temperature rises (by a maximum of 0.5 °C) in night-time when compared to smaller trees. The spacing of trees also plays a significant role in modifying and cooling the environment: a 6 m tree spacing had a better impact than an 8 m spacing, regardless of the cost of maintenance, required irrigation systems, and pricing of the materials [25].

Finally, Wang et al. attempted to analyze the impact of one and two trees on the airflow using the OPEN FOAM tool in street canyons with different aspect ratios. Generally, tree branches modify the air flow speed and direction, which was especially witnessed in the two-trees case. In a porous area, the air flows easily and slowly, depending on the spacing between trees [26].

*1.4. Problem Statement*

Egypt—especially the Greater Cairo Region—has witnessed recent urban vehicular network re-development that would lead one to expect a change in the microclimate due to the need to eradicate parts of the green fabric in some streets. As previously mentioned, Cairo, a metropolis city, is already witnessing an increased UHI and high range of greenhouse gas emissions due to over-population, traffic emissions, and urban emissions, among other aspects. It is expected that these impacts will be doubled, affecting environmental, economic, and social aspects. However, Green Infrastructure—and urban trees in particular—can enhance the microclimate of spaces, through solar irradiance reduction under its canopy. This area of research is currently under widespread scrutiny, and investigation is required to guide planners to develop optimal tree spacing and arrangement designs when such urban vehicular re-development is performed.

Therefore, this research focuses on an urban street in one of the eastern Cairo neighborhoods, named El-Nozha district. The parameters Air Temperature ($T_a$), Mean Radiant Temperature ($T_{mrt}$), Relative Humidity (RH), Wind speed (V), and Physiological Equivalent Temperature (PET) were calculated before and after introducing urban trees with different arrangements and spatial locations, through the use of parametric simulation. The proposal is solely focused on urban trees, as they have a prime impact on the urban environment, in comparison with other vegetative typologies. In this investigation, we propose a novel method to quantify the relationship between metrological parameters along the day for mid-rise compact urban streets.

## 2. Methodology

*2.1. Case Study*

The urban street, El Shaheed Sayed Zakaria, is located at 30°06′ N and 31°22′ E, in the Eastern part of Cairo (El-Nozha district). The street is in the heart of the Sheraton Heliopolis area, an urban residential district recognized for its open green spaces. Recently, it has witnessed a great urban transformation undertaken to facilitate the urban mobility of the area; the middle green island has shrunk to 22 m wide, including sidewalks, and the asphalt road has been widened to 26 m, including two-sided parking areas. This urban street was specifically selected as a case study, as most of the re-development has been undertaken in similar areas (i.e., mid-rise compact developments). A street sector approx. 1 km and 100 m wide is investigated in this study, which is indicated in Figure 2.

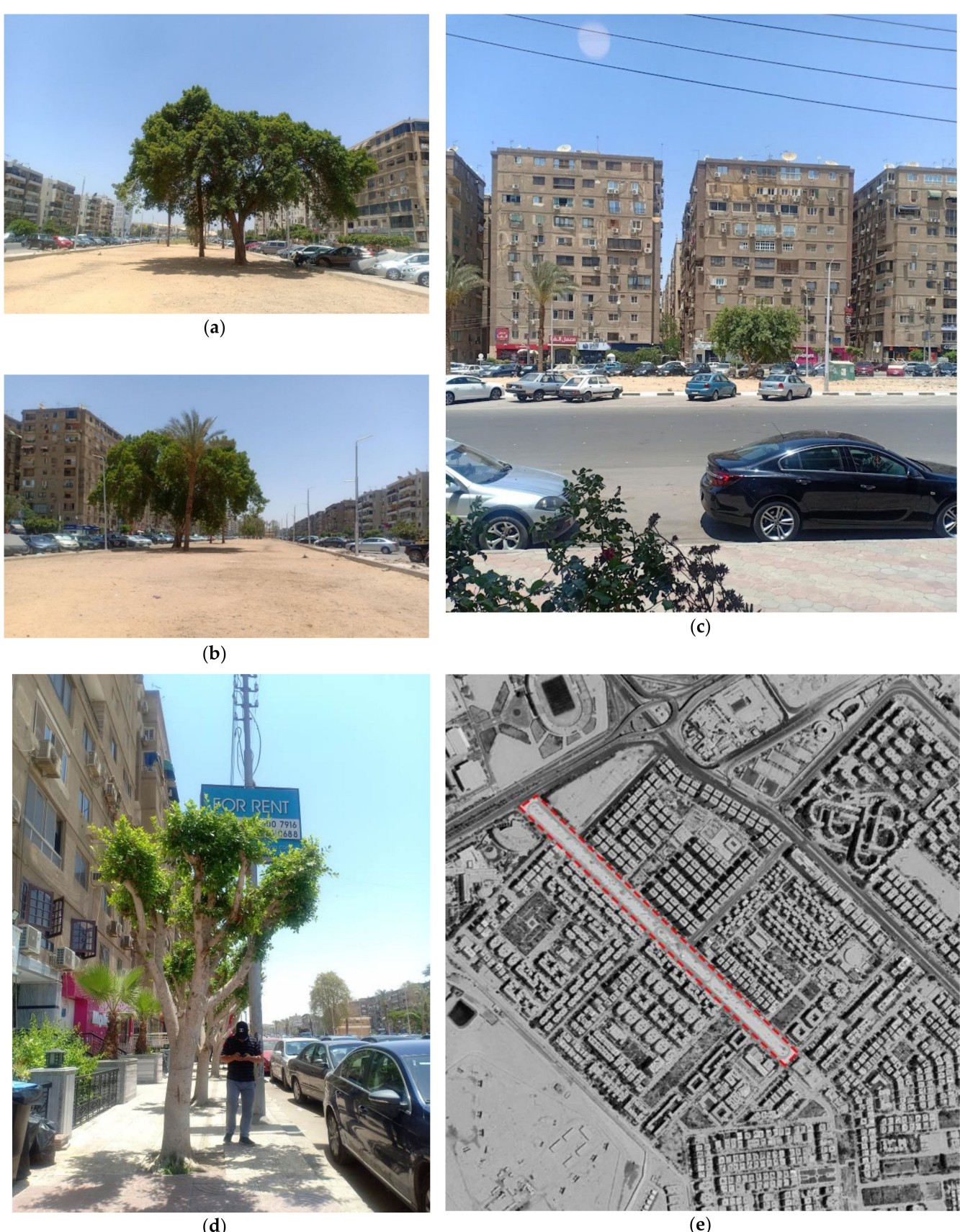

**Figure 2.** El Shaheed Sayed Zakareya street: (**a**–**c**) Different views for the whole street, including the middle garden, parking area, and asphalt road; (**d**) sidewalk urban trees located in the street, both photos are captured on the 10th of June 2021; and (**e**) satellite imagery of the studied street sector using Google Earth, captured on 28th May 2021.

Most of the buildings are mid-rise residential buildings with commercial use on the ground floor, with heights ranging from 6 to 11 floors. The street section shown in Figure 3a elaborates the current surface materials (asphalt surface, yellow tiles, and concrete surfaces), illustrating the newly developed street design section. Figure 3b gives an overview of the percentage of each surface material, including the green area. The green area, placed in the middle of the street, covers approximately 24,440 m$^2$, accounting for 17% of the total street area. The present trees are around 2% of the total area, spatially located inside the middle green island and the sidewalks next to the buildings. Details of the existing tree stock and the plantation pattern are shown in Table 1.

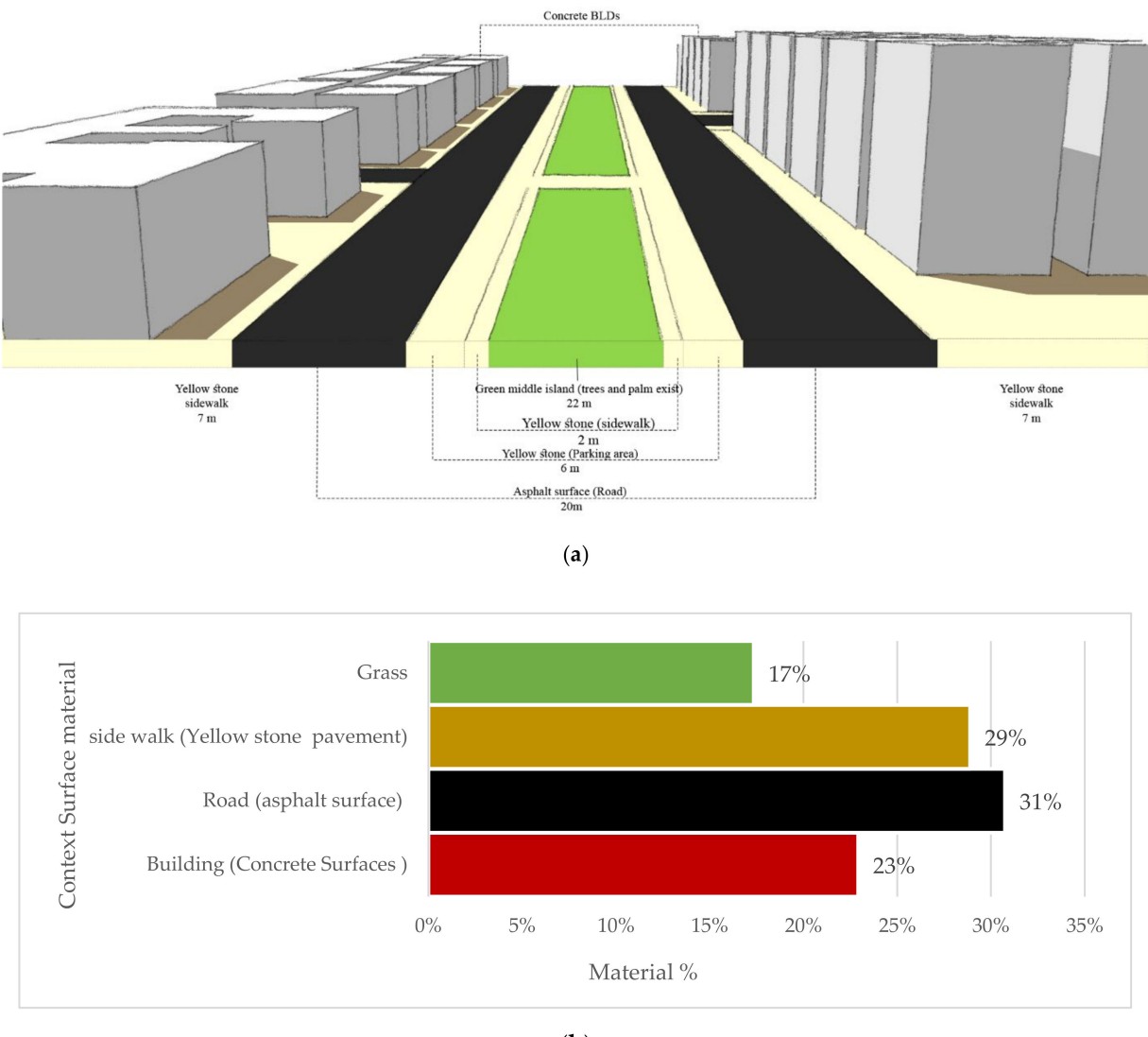

(a)

(b)

**Figure 3.** (**a**) Street cross-section elaborating the existing materials of the context; and (**b**) the percentage of surface material existing in ElShaheed Sayed Zakareya Street (authors' elaboration).

**Table 1.** The existing tree stock and plantation patterns.

| Tree | No. of Trees | | | | Canopy | | | |
|---|---|---|---|---|---|---|---|---|
| Tree Details and Code | East Side | Middle Island | West Side | Total No. | Coverage (m²) | % | Albedo | Shape and Size |
| Evergreen (small), SS | 58 | 0 | 18 | 76 | 532 | 12% | 0.12 | Conic, small trunk, sparse, small (5 m) |
| Palm (large), DM | 0 | 26 | 2 | 28 | 1781.36 | 41% | 0.18 | Palm |
| Evergreen (medium), SM | 0 | 6 | 21 | 27 | 1717.74 | 40% | 0.12 | Conic, large trunk, sparse, medium (15 m) |
| Evergreen (large), SL | 0 | 2 | 0 | 2 | 190.06 | 4% | 0.12 | Spherical, large trunk, sparse, large (25 m) |
| Deciduous dense (small), DS | 0 | 0 | 18 | 18 | 126 | 3% | 0.18 | Spherical, small trunk, dense, small (5 m) |

Notes: Total no. of trees = 151; total coverage of trees = 2% of the total case study area. Since the site is still in the construction phase, for ENVI-met simulation, all trees were considered healthy, receive regular maintenance, are fully furnished with lawns, and are well-irrigated. The tree albedo was adopted from the Albero 4.4.3 modeling tool inside the ENVI-met software, as the LI-COR plant canopy analyzer was not available to use.

The climate of Cairo, based on the Köppen Climate Classification, is a hot arid climate (BWh) climatic zone. Using the weather file monitored at Cairo International Airport [27], the average temperature in a typical summer week is 27.5 °C with 54% relative humidity, while it decreases to 15.5 °C in a typical winter week, with 62% relative humidity. During extreme heat waves, the temperature certainly fluctuates to higher degrees. The average wind speed throughout the year experiences a mild variation. From mid-July to early March, the wind records the calmest speed (3.5 m/s), while it is windier from early March to mid-July, with records higher than 4 m/s. The dominant wind comes from the North, then Northeast and Northwest, with averaged wind speeds of 3.79, 4.19, and 3.56 m/s, respectively. The albedo of the surfaces and the urban morphology of the context strongly affects the temperature ranges and the wind pattern [28].

### 2.2. Methods and Scenarios

Four methods for conducting the study were used: (1) on-site mapping of the currently existing urban trees, investigating the surface materials present in the street using the author's knowledge and expert consultation, and surveying the urban geometry; (2) local climate prediction using the Meteonorm tool; (3) validation using hourly on-site field measurements and validation of computer-based simulation (through the ENVI-met software); and (4) finally, simulation and analysis of results of the different developed tree patterns (see Figure 4).

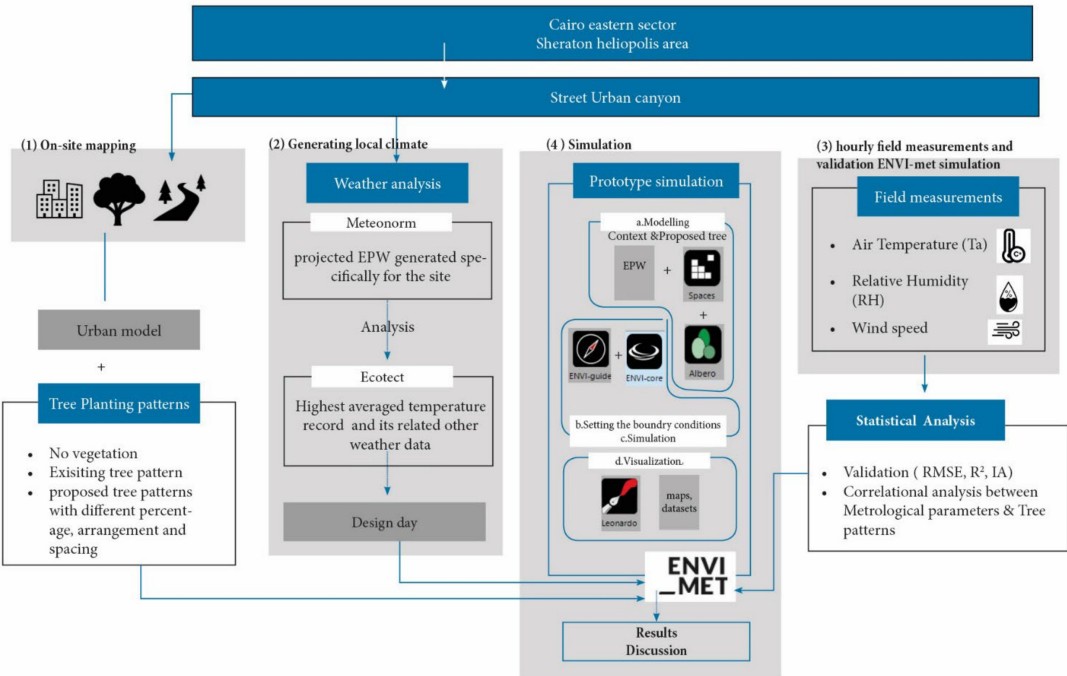

**Figure 4.** Methodology used within this study.

### 2.2.1. Urban Trees

The unavailability of the LI-Cor plant canopy analyzer limited the modeling of existing trees; thus, the built-in tree database in ENVI-met was utilized. Trees with similar geometry and foliage pattern were selected; the existing tree details are formerly mentioned in Table 1 (in the case study section). Additionally, for the developed plantation scenarios, we proposed the use of *Cassia nodosa*. This tree is commonly found in Cairo, as it is well-adapted to hot climates, it is a deciduous tree frequently used in urban parks, and it provides sufficient shade in the summer and heat in the winter, when its leaves fall [29]. The detailed inputs required in the ENVI-met tool, such as the Leaf Area Index (LAI) and albedo, were adapted from the measured values given in [30]; see Table 2. This tree was modeled and added to ENVI-met 3D plants database, following the work of Fahmy et al. [26,27] using the Albero tool inside the ENVI-met software.

**Table 2.** Proposed tree specification. LAI was adopted from [30], geometry gathered from [30], and LAD values were calculated based on [31] for *Cassia nodosa*.

| Specification | Proposed Tree |
|---|---|
| Name | *Cassia nodosa* |
| Alternative Name | Pink shower |
| Foliage profile based on [30], analyzed by LAI2200 data analyzer | |
| Total tree height | 7 |
| Maximum LAD at height | 5 |
| Foliage height | 3 |
| Foliage Albedo | 0.18 |
| LAI Leaf Area Index | 1.61 |
| LAD Leaf Area Density | |
| 1 m | 0 |
| 2 m | 0 |
| 3 m | 0.189712729 |
| 4 m | 0.392133918 |
| 5 m | 0.604495398 |
| 6 m | 0.518514321 |
| 7 m | 0 |

### 2.2.2. ENVI-Met Modeling Scenarios, Calibration, and Sensitivity

The purpose of the calibration and validation processes that took place was to ensure the sensitivity and suitability of data entry (including meteorology and modeling parameters), as well as the output parameters. Considering this, all data were calibrated to reality using a well-established modeling tool (Spaces) integrated in the ENVI-met software, using 3D scaling for X, Y, and Z from 1 to 10. On the other hand, we carried out previous simulation experiments that demonstrated the suitability of ENVI-met modeling and data entry for further experiments [9,30,32].

The calibration of the model went through two stages: first, we calibrated the needed meteorological data set for the street 30°06′ N and 31°22′, which was generated by the Meteonorm tool [33], and analyzed and visualized using the Ecotect tool [34], as shown in Figure 5. The tool generates Typical Meteorological Year (TMY) weather data for the required location, based on field measurements from the nearest weather stations, which is accepted as long as it is within no more than 3 km, according to the WMO measurement instructions [35]. Meteonorm creates hourly time-step weather files in various formats

(i.e., EPW and STAT files), that could be refined or coupled with other simulation tools, consistent with accepted accuracy [22,36–41]. The 20th of July was chosen as the simulation day, as it represented both the peak and typical hottest days of summer in the examined site, as shown in Figure 5.

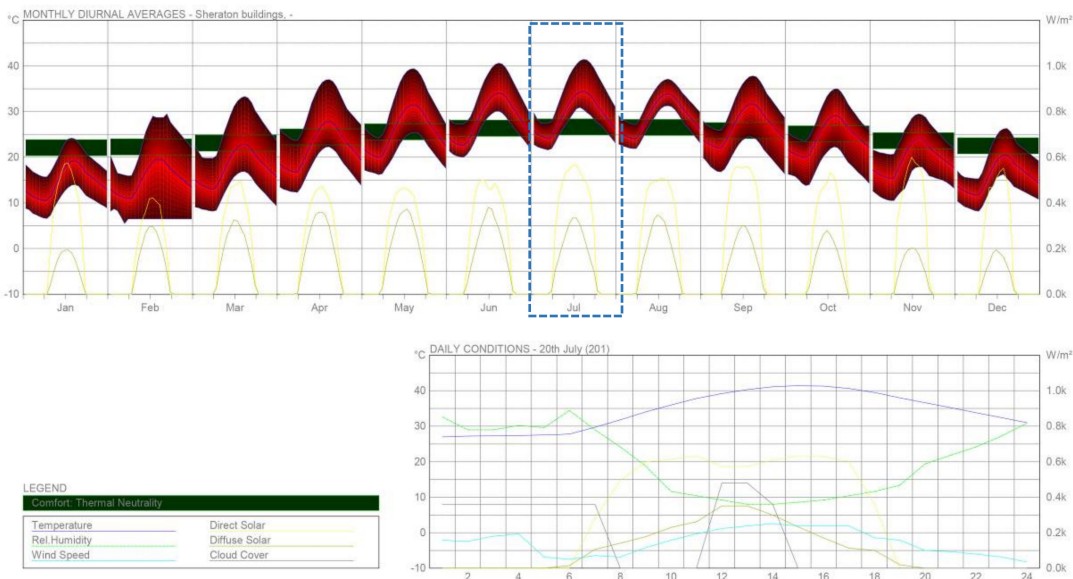

**Figure 5.** Monthly averages and hourly data for 20th of July, as it is the peak average hottest day, analyzed using Ecotect 2010 [34] tool.

Second, we calibrated the built environment element geometry and the model boundary conditions. Built environment elements, including buildings, vegetation, and street networks, were modeled using the definite measured as-built geometry, materials, and vegetation types expressed in the model databases and in Section 2.2.1, for urban trees with typical scaling in the three dimensions X, Y, and Z; see Table 2. The East–West axis is expressed by the X dimension, the North–South direction is expressed by the Y dimension, and Z denotes the height of the buildings. Boundary conditions are described, in ENVI-met, by metrological data entry through its configuration wizard tool, where the TMY data extracted in the first stage of calibration can be entered to express all urban canopy layer microclimatic conditions of the site.

The site was downscaled into a 42 × 191 (dx = dy = 5) grid horizontally, and a 12 × (dz = 3) grid vertically, with three nested grids to ease the air flow and wind turbulence calculations. Asphalt surfaces and yellow tiles were kept as the existing case, and sandy soil was considered the base soil structure; see Table 3. Nine different urban simulations were developed, as shown in Table 4. A couple of previous studies have mentioned research performed by selecting suitable tree species, arrangements, and sizes. Thus, for this study, we systematically designed the following scenarios:

- Random arrangement was eliminated from the study, as it ameliorates the climate less when compared to clustered and equidistant arrangements. This finding has been previously discussed by Zhao et al. and Amani-Beni et al. [24,39].
- In the designed scenarios, the spacing between trees was 5 m and equidistant, as Abu Ali et al. have shown that 6 m spacing performs better than 8 m, with respect to climate moderation. A 5 m spacing was selected, according to the developed horizontal grid (dx = dy = 5) for the site ENVI-met model.
- Following the previous point, no spacing was added to the scenarios, in order to maximize the tree density with no overlap. The superior performance of small, dense trees has been discussed by Zhao et al. (2008) [24].

- Small-sized trees were selected in the first place, knowing that their performance for moderating temperature is less than large trees during the day; however, at night, the smallest temperature increase has been observed with small trees [16].

**Table 3.** Input data for simulation.

| Input Parameter | Value(s) Used |
|---|---|
| City location | Cairo, Egypt (Lat: 30.06°; Long: 31.25°) |
| Simulation day | 20th July, 2020 |
| Simulation duration | 24 h |
| Model resolution | $42 \times 191 \times 12$ m |
| Climate system | Hot arid |
| Windspeed and direction | 3 m/s at 240° |
| Temperature | Min: 26.4 °C; Max: 39.70 °C (forced sampling) |
| Relative humidity | Min: 27%; Max: 59% |
| Cloud cover | Default setting in ENVI-met |
| Material albedo | Grass = 0.2; yellow tiles = 0.5; asphalt = 0.2 |
| Soil | Sandy soil (physical properties based on ENVI-met database) |
| Building wall and roof materials | Default settings in ENVI-met |

Note: The albedo of the materials was adopted from the ENVI-met model built-in library, as on-site measurement was not possible for the authors, due to a lack of measurement tools.

**Table 4.** Scenarios generated for optimization.

| Green Design Scenario | Description | Vegetative Share (%) (inc. (a) Grass Area. (b) Canopy Coverage) | Number of Trees | Visualization |
|---|---|---|---|---|
| Sc_0 | Current greening pattern within the street. | (a) 17% (b) 2% | 151 | |
| No_Veg | No vegetation | 0% | 0 | |
| Sc_1 | Single-side trees with 5 m spacing | (a) 17% (b) 3% | 224 | |
| Sc_2 | Single centered tree with 5 m spacing | (a) 17% (b) 2% | 112 | |
| Sc_3 | Clustered trees on side | (a) 17% (b) 4% | 252 | |
| Sc_4 | Clustered trees centered | (a) 17% (b) 2% | 126 | |
| Sc_5 | Side staggered clustered trees | (a) 17% (b) 3% | 231 | |
| Sc_6 | Side clustered trees and single tree centered | (a) 17% (b) 4% | 315 | |
| Sc_7 | Single-side trees clustered center trees | (a) 17% (b) 4% | 252 | |

Hence, the proposed scenarios varied, in terms of: (1) The location of the tree in the island—in the middle of the island or on its sides; (2) clustered/grouped trees—either the trees are standalone, or placed in pairs without canopy overlap; and (3) hybrid model—varying both the location and the tree clusters. Assessment was conducted at 11:00, 16:00, and 19:00 for six receptor points at a level of 1.5 m above the ground, as this is approximately the mean height of a human head or neck [42]. Receptors were placed in an equidistant manner: R1, R3, R4, and R6 were placed on the building sides, to quantify the impact of middle island plantation patterns, while R2 and R5 were placed inside the green island, in order to quantify the direct impact of the proposed designed scenarios (see Figure 6).

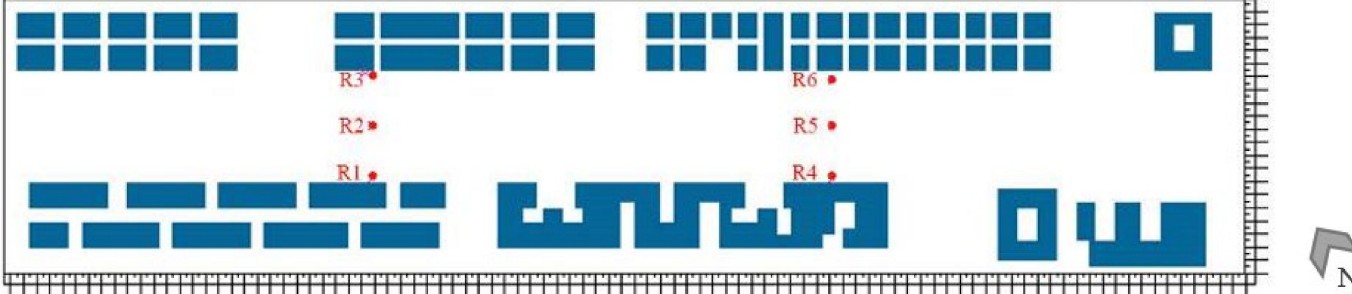

**Figure 6.** Receptor locations for assessment.

### 2.2.3. ENVI-Met Model Validation

ENVI-met is a three-dimensional microclimate simulation model that simulates air–plant–building–soil interactions in built environments [43]. The model considers all the vegetation aspects, as it simulates and calculates the heat and vapor exchange between the plants and their surroundings. It also simulates the evapotranspiration cycle from the roots to stomata of the leaves, photosynthesis, and the leaf temperature of the tree. The 3D vegetative grid-based simulation model named Albero in ENVI-met software was developed to model complex geometry. Details are required for tree modeling (i.e., Leaf Area Density, Leaf Area Index, geometry of the tree, the foliage albedo, and the depth and the diameter of the roots) [43,44]. The outcomes for physical properties such as ET and shade are sufficient and reliable, according to previous studies [9,30,32]. The model output has been validated in various research (i.e., [30,38]) and its reliability has been proven [44]. This research attempts to validate ENVI-met simulation results. Thus, hourly field measurements were taken for three microclimatic variables: Air temperature ($T_a$), relative humidity (RH), and wind speed (V) on the street (Lat: 30.06°; Long: 31.25°) on the 28th of June 2021. Air temperature and relative humidity were recorded using a Portable TroTec Bc 20 [45] at 1.5 m (above the ground), the approximate level of human breathing [42,46] and a wind speed Techno Line ea3000 [47]; details of the equipment used are listed in Table 5.

**Table 5.** Measuring devices used to measure air temperature, humidity, and wind speed.

| Measurement Instrument | Measuring Parameter | Accuracy | Photo |
|---|---|---|---|
| Portable TroTec Bc 20 | Air Temperature Relative humidity | $\pm 1\,°C$ $\pm 2\%$ | |
| Techno Line ea3000 | Wind speed | $\pm 5\%$ | |

The predicted outputs of ENVI-met and on-site measurements were compared, as shown in Table 6 and Figure 7, through use of the Pearson's coefficient of determination ($R^2$), Root Mean Squared Error (RMSE), and the Index of Agreement (IA). Not all of the needed meteorological data for ENVI-met were measured, due to site conditions (heat wave and construction dust); only the three previously mentioned parameters were measured. Values close to 1 are desirable for $R^2$ [48] and IA [49], whereas a lower value for the RMSE is better, as it expresses the standard deviation of the difference between simulated and measured values [50].

**Table 6.** Error calculations comparing values predicted by ENVI-met and on-site measured values.

| Parameter/Error | $R^2$ | IA | RMSE |
|---|---|---|---|
| Wind Speed (m/s) | 0.6957 | 0.177 | 0.4 |
| Air Temperature (°C) | 0.8946 | 0.5 | 5.6 |
| Relative Humidity (%) | 0.971 | 0.839 | 8.5 |

As shown in Table 6, generally, the statistical validation showed good agreement between measured parameters and the ENVI-met simulation output. $R^2$ and IA for Air temperature and Relative humidity values indicated a prominent level of agreement between the results. $T_a$ and RH showed strong correlations, with $R^2$ values of 0.8946 and 0.971, and IA values of 0.5 and 0.839, respectively. By contrast, the wind speed obtained a moderate correlation ($R^2$ of 0.6957, RMSE of 0.4, and no agreement with IA). The significance calculations witnessed for the RMSE of $T_a$ and RH and the lack of agreement for the wind speed can be attributed to the site conditions during measurements. Site constructions were present, and a non-typical heatwave was witnessed, recording 41.08 °C at 14:00 LST on the 28th of June. The insignificance of the wind speed parameters ($R^2$, IA, and RMSE) can be attributed to the non-windy conditions on the study day, which caused most of the measurements to be no more than 0.5 m/s, as shown in Figure 7a; hence, it can be argued that these insignificant calculations for the wind speed were not reliable. However, the findings presented in this section are remarkably close to those found in the works of Shata et al. [31] and Elwy et al. [41], who determined the insignificance of the wind simulation results at 0.5 m/s and below, which might require ENVI-met to allow 24 h wind speed data entry, as it accepts 24 h data for $T_a$ and RH. Such calibration detail was treated by using meteorological data for the examined site from Meteonorm, as described in Section 2.2.2.

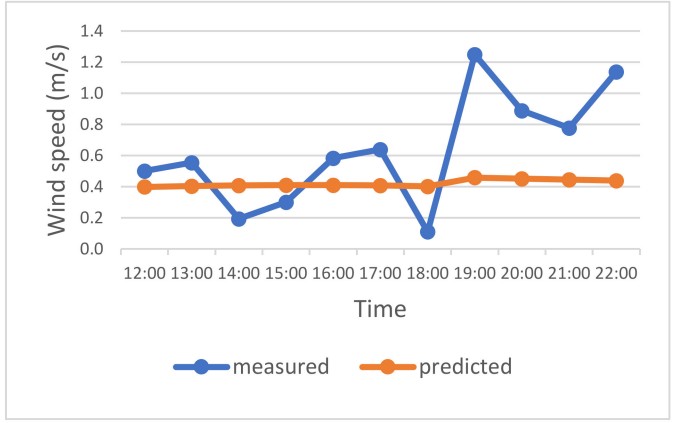

(a) Measured and predicted values of the wind speed

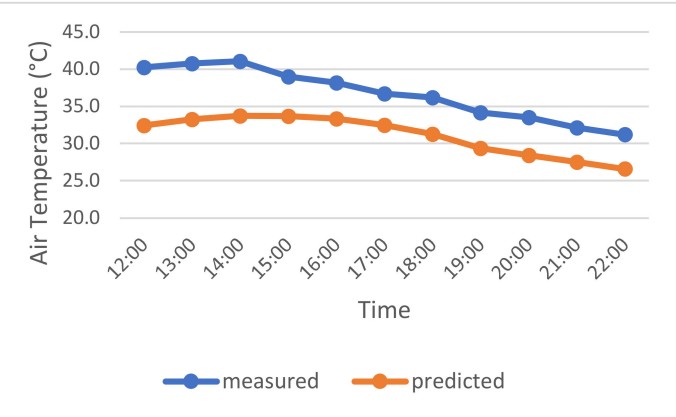

(b) Measured and predicted values of Air temperature

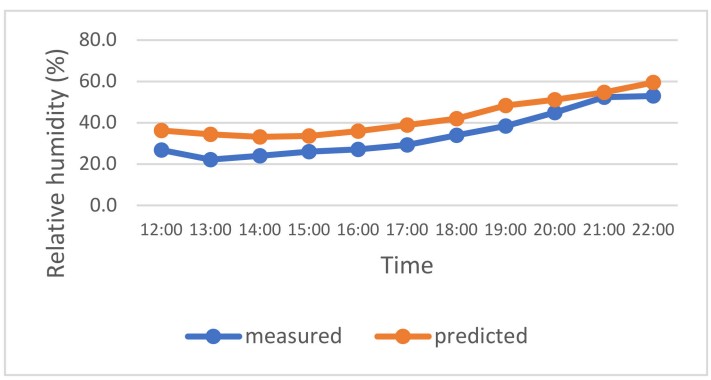

(c)   Measured and predicted values of the Relative humidity

**Figure 7.** Comparisons of (**a**) wind speed (m/s); (**b**) Air temperature (°C); and (**c**) Relative Humidity (%), between ENVI-met and field measurements.

## 3. Results

This section begins with a comparison of the current vegetative pattern (including grass as a ground cover) and the no-vegetation scenario, in order to assess and quantify the cooling effect and the shade flux that the existing trees provide. Assessments were conducted at 11:00, 16:00, and 19:00 for six receptor points 1.5 m above the ground, to quantify the direct impact of the proposed scenarios. Then, comparative analysis was performed between the different designed vegetative patterns and the no-vegetation case. Finally, correlational analysis is introduced between climatic parameters, before and after proposing the use of vegetation.

### 3.1. Current Design and No-Vegetation Scenarios

To quantify the cooling effect of vegetation in general, the existing design case was compared with the no-vegetation case. The current design condition was simulated as: (1) The middle island fully covered with grass; (2) the present trees and palms in the middle island; (3) side trees next to the buildings are considered; and, finally, (4) the sidewalks and pavements are the same as in the no-vegetation scenario.

Through the results shown in Figure 8 and Table A1 in Appendix A, the difference in the overall maximum temperature range between the vegetation and no-vegetation scenarios was 0.01 and 0.08 °C at 11:00 and 16:00, respectively, and no change was observed at 19:00. Even though a temperature difference was clearly noted in the overall minimum range at 11:00, the temperature was reduced by 0.76 °C, a slight improvement was observed

at 16:00 (equal to 0.08 °C), and a minor temperature shift (by 0.07 °C) was observed when it reached 19:00.

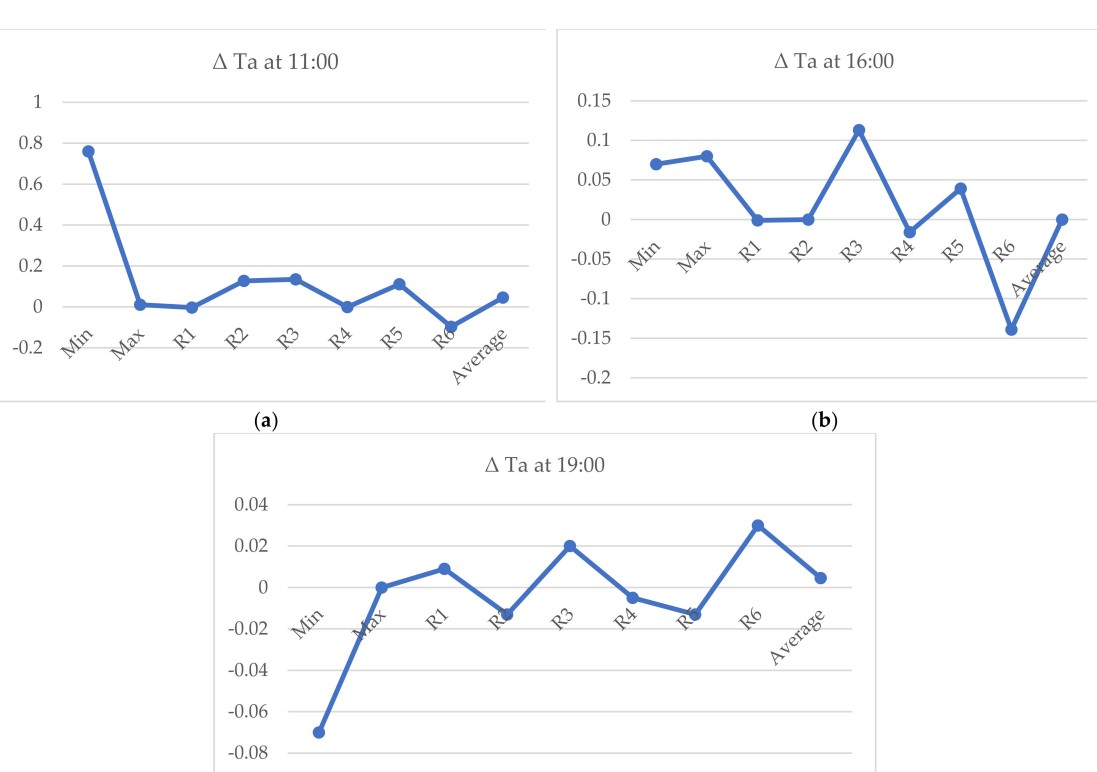

**Figure 8.** The temperature differences between the no-vegetation and current plantation patterns: (**a**) Temperature difference at 11:00; (**b**) temperature difference at 16:00; and (**c**) temperature difference at 19:00.

At the receptor points, the urban morphology and the spatial placement of trees strongly affected the temperature ranges [51]. Receptors 1 and 4, placed on the west of the island next to the buildings, generally recorded minor increases in temperature in the presence of vegetation at 11:00 and 16:00. The highest temperature difference was at Receptor 4, at 16:00, which recorded a difference of 0.016 °C. Temperature in the existing design was 38.523 °C, this record is higher than in the no-vegetation scenario. An inverse finding was clearly seen at Receptor 3, located to the east of the island next to the buildings, where the temperature was reduced by 0.135, 0.113, and 0.02 °C at 11:00, 16:00, and 19:00, respectively, in the presence of vegetation. Meanwhile, the lower Receptor 6 recorded higher temperature ranges at 11:00 and 16:00, reaching differences of 0.097 and 0.139 °C, respectively; whereas at night, the temperature decreased minorly by 0.0046 °C.

Within the street island, at Receptors 2 and 5, the temperature difference fluctuated positively when vegetation was present, except at 19:00. At 11:00, the highest difference at both receptors was observed (0.127 and 0.11 °C, respectively, whereas at 16:00, the differences were 0 and 0.039 °C, respectively). It is important to note that, on the west side of the island, the canopy coverage is higher than on the east side of the island. Small evergreen trees are present in the eastern part; however, the size of the trees varies, as mentioned in Table 1.

Simulation of the existing vegetative pattern and the no-vegetation pattern showed that the presence of trees and grass enhanced the overall temperature by up to 0.76 °C, due to the tree canopy shading flux and evapotranspiration effects [32]; however, at night, the temperature minorly increased, as the heat and humidity trapped under the tree canopy is released, causing a warmer pattern [21]. Considering the receptor points, it is clear that the receptors under and beyond the tree canopy recorded a better temperature range, as witnessed in the middle island, except at night. Meanwhile, the temperature varied at

receptors next to the buildings, as the wind was modified and the humidity was elevated, due to the increase in foliage range [9] and the small-sized canopy, in some cases.

### 3.2. Results of the Proposed Scenarios

The results of the proposed scenarios fluctuated, according to the timing along the day. Common rules were applied to all of the scenarios: (1) A deciduous medium-sized tree was used and grass was the ground cover; (2) trees were placed only in the middle island, and no trees existed on the sides of buildings; and (3) the present trees were removed to quantify the proposed tree effects, as well as the records of each receptor and the averaged values.

The mean radiant temperature visualized maps are provided in Appendix B (Figures A1–A3), and the temperature records under the different plantation designs are given in Appendix A (Table A2). These scenarios were compared with the no-vegetation case, in terms of the differences at 11:00, 16:00, and 19:00, as graphically presented in Figure 9.

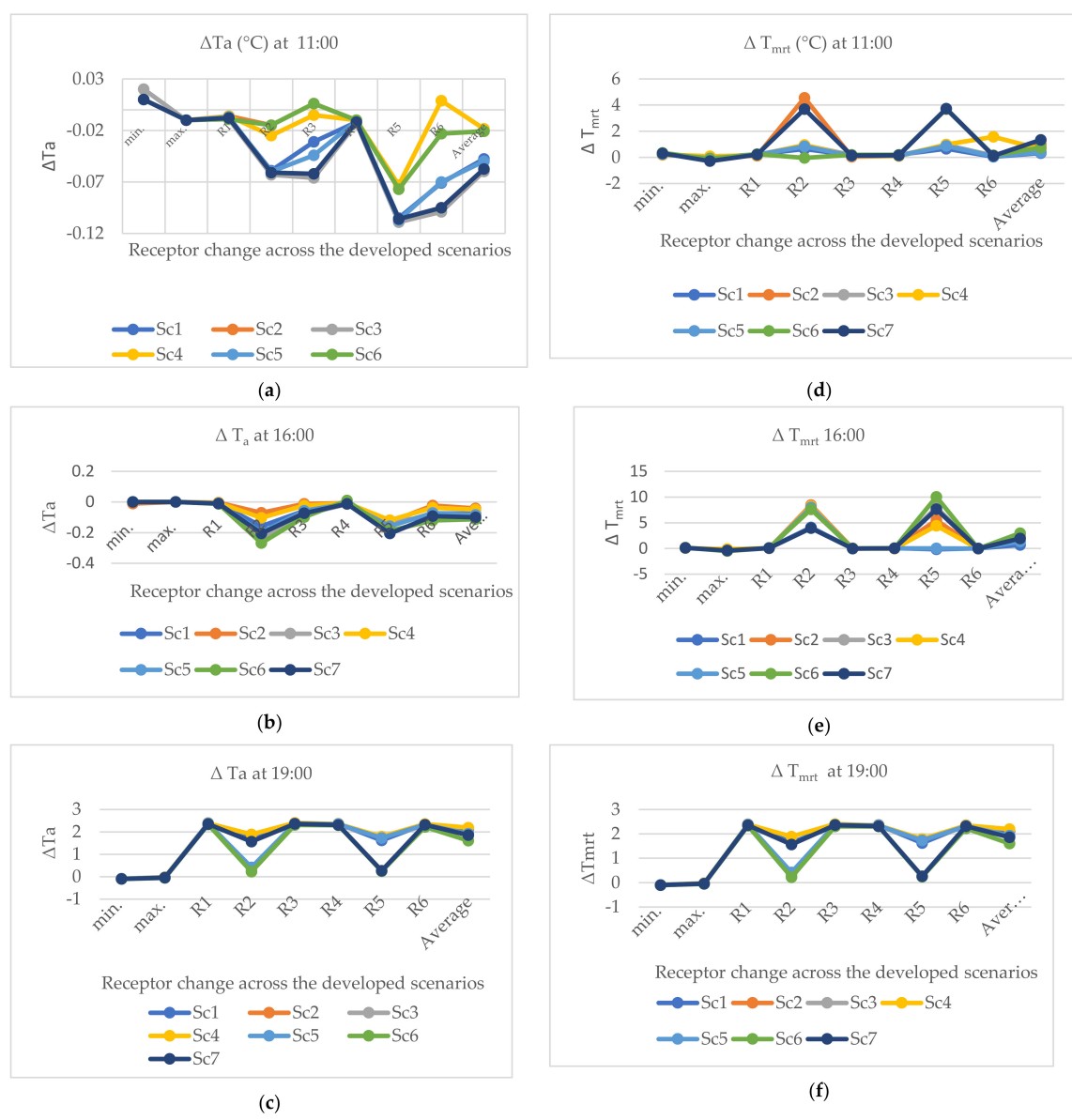

**Figure 9.** Temperature difference between the no-vegetation and proposed scenarios: (**a**) Temperature difference at 11:00; (**b**) temperature difference at 16:00; (**c**) temperature difference at 19:00; (**d**) mean radiant temperature at 11:00; (**e**) mean radiant temperature difference at 16:00; and (**f**) mean temperature difference at 19:00.

Temporal Responses to Trees Coverage

- At 11:00

    In all scenarios, we clearly witnessed that the overall minimum temperature was reduced by less than 0.02 °C, except with the clustered trees on the side arrangement, where the reduction reached 0.02 °C. By contrast, the maximum temperature in all of the scenarios was minorly elevated by less than 0.02 °C. The same finding was clearly seen for the receptor points in all scenarios, except for the centered trees and side clustered trees with center trees arrangement scenarios at Receptor 3, where the temperature was reduced by 0.006 °C. At Receptor 6, the temperature was reduced by 0.01 °C with trees clustered in the center.

    Differing from the temperature records, the presence of trees clearly reduced the overall minimum, maximum, and all receptors' simulated $T_{mrt}$. All the scenarios, except side trees and clustered centered trees, showed increased values in the overall maximum $T_{mrt}$ values by a maximum of 0.29 °C when trees were clustered in the center with side arrangement and a minimum of 0.01 °C when trees were single-side arranged.

    As Receptors 2 and 5 were placed in the middle island, the major reduction was witnessed, reaching 0.636, 4.551, 0.927, 0.957, 0.84, −0.036, and 3.693 °C at Receptor 2 and reaching 0.663, 0.923, 0.954, 0.988, 0.869, 3.696, and 3.725 °C at Receptor 5 for the side trees, centered trees, side clustered trees, centered clustered trees, side staggered clustered trees, side clustered with single centered trees, and single-side with clustered trees scenarios, respectively. A clear decrease was seen in Receptor 6 under the clustered center trees scenario, reaching 1.569° C, whereas that for the rest of the scenarios ranged from 0.05 to 0.163 °C. Other receptor points, such as 1, 3, and 4, encountered a minor reduction in all scenarios. In the averaged records, the single-side with clustered center trees scenario recorded the highest overall reduction (1.342 °C), while the lowest difference (in the single-side trees scenario) was equal to 0.32 °C.

- At 16:00

    Nearly no change in the overall minimum and maximum temperature readings was observed. The implementation of vegetation in all scenarios negatively affected the temperature at all receptor points in varying ranges, as grouped trees can significantly reduce the Sky View Factor (SVF) during daytime and as the tapped heat slowly releases during the night, promoting higher humidity levels in the context [16]. The highest increase in temperature was 0.26 °C, at the upper middle receptor, when trees were clustered on the side with single centered trees. Receptors located at the east of the buildings (1 and 4) recorded the lowest temperature change in all scenarios. The smallest averaged temperature receptor differences were under the centered tree and clustered centered trees arrangements (0.041 and 0.05° C, respectively). Meanwhile, the highest readings were 0.1 °C, when the tree arrangements were side clustered with single centered trees and its inverse, and a difference of 0.09 °C was observed with clustered trees arranged on the side.

    There was a minor change in radiant temperature values in the overall minimum records, as well in the side receptors 1, 3, 4, and 6. The change was solely observed in the middle island, due to the presence of trees with different arrangements. Sun shield enhancement reached a 8.5 °C maximum and a 4 °C minimum in Receptor 2, whilst a maximum of 10 °C and a minimum of 0 °C were observed for Receptor 5. The reduction under side clustered with centered trees and clustered trees arrangements outweighed those of single centered tree and side staggered clustered tree arrangements (i.e., the lowest records in scenarios 1 and 5 were 0.7 and 1.34 °C, respectively).

- At 19:00

    Finally, the overall minimum and maximum temperature were increased by a maximum of 0.1 °C in all scenarios; however, reduction was clearly seen in the different receptors. Under the tree canopy at the middle receptor (r5), the reduction ranged from 1.6 to 1.8 °C for all scenarios, except the scenario of side clustered with single centered trees

and its inverse, when it reached 0.3 °C. Conversely, the single-side trees with clustered trees in the center arrangement reduced the temperature to nearly 1.5 °C at Receptor 2, and single centered trees and clustered side trees decreased the difference by 1.5 and 1.3 °C, respectively, compared to Receptor 5. Generally, the average records of the six receptors in all scenarios showed temperature reduction; the best scenarios were the single-side trees and clustered centered tree arrangements, recording the highest differences of 2.2 and 2.1 °C, respectively, while the single centered tree and clustered side tree arrangements were nearly the same, with a 1.9° C reduction. The lowest temperature difference recorded (1.6 °C) was for the side clustered with single centered trees arrangement.

The overall change in $T_{mrt}$ was negligible in all the scenarios. The receptor points provided a better illustration for the night-time records: Side receptor (1, 3, 4, and 6) reductions ranged between 2.2 and 2.4 °C, while that of the middle receptor (r5) ranged between 1.6 and 1.8 °C, except under the side staggered clustered trees and the side clustered with single centered trees arrangements (0 and 2 °C, respectively). The results for the upper middle receptor (r2) fluctuated between the scenarios: The centered cluster trees, single-side trees, and single-side and cluster center trees scenarios recorded a higher range than the rest, with a maximum of 1.9 °C and minimum of 0.2 °C.

### 3.3. Correlation of Meteorology and Pedestrian Thermal Comfort

The association with pedestrian thermal comfort could best be determined through correlational analysis between metrological variables—Air Temperature ($T_a$), and Mean Radiant Temperature ($T_{mrt}$)—with and without the presence of vegetation. All simulation outcomes, for a total of 162 records, were entered into a Microsoft excel spreadsheet, for analysis using Pearson's correlation and the coefficient of determination ($r^2$) [52]. It is expected that the outcome of these correlations can be used in case of the same climatic conditions as stated in Section 2.2.2 (ENVI-met validation and calibration).

The no-vegetation scenario accounted for 18 records, at 11:00, 16:00, and 19:00 for the six receptor points. The eight scenarios, including the different developed tree patterns and the existing case, accounted for 144 records. The findings were visualized through correlational matrices, as seen in Tables 7 and 8, while scatterplots are shown in Figures 10–12.

**Table 7.** Correlational matrix between climatic parameters in the no-vegetation case.

|  |  | PET (°C) | $T_a$ (°C) | $T_{mrt}$ (°C) | RH (%) |
|---|---|---|---|---|---|
| 11:00 | PET (°C) | 1 | −0.02 | −0.88 | −0.24 |
|  | V (m/s) | −0.95 | −0.21 | 0.97 | 0.46 |
|  | $T_a$ (°C) | −0.02 | 1.00 | −0.28 | −0.95 |
|  | $T_{mrt}$ (°C) | −0.88 | −0.28 | 1.00 | 0.50 |
| 16:00 | PET (°C) | 1 | 0.93 | 0.90 | −0.91 |
|  | V (m/s) | 0.26 | −0.09 | 0.65 | 0.16 |
|  | $T_a$ (°C) | 0.93 | 1.00 | 0.69 | −1.00 |
|  | $T_{mrt}$ (°C) | 0.90 | 0.69 | 1.00 | −0.64 |
| 19:00 | PET (°C) | 1 | 0.48 | 0.56 | −0.70 |
|  | V (m/s) | 0.54 | −0.48 | 0.71 | 0.22 |
|  | $T_a$ (°C) | 0.48 | 1.00 | −0.16 | −0.95 |
|  | $T_{mrt}$ (°C) | 0.56 | −0.16 | 1.00 | −0.11 |
| Strong correlation |  | Moderate correlation |  | Same parameters |  |

**Table 8.** Correlation between climatic parameters in the case of different vegetative patterns.

| | | PET (°C) | $T_a$ (°C) | $T_{mrt}$ (°C) | RH (%) |
|---|---|---|---|---|---|
| **11:00** | PET (°C) | 1.00 | −0.14 | 0.54 | −0.21 |
| | V (m/s) | −0.66 | −0.19 | 0.24 | 0.50 |
| | $T_a$ (°C) | −0.14 | 1.00 | −0.22 | −0.90 |
| | $T_{mrt}$ (°C) | 0.54 | −0.22 | 1.00 | 0.12 |
| **16:00** | PET (°C) | 1.00 | 0.84 | 0.94 | −0.89 |
| | V (m/s) | 0.19 | 0.07 | 0.52 | 0.00 |
| | $T_a$ (°C) | 0.84 | 1.00 | 0.75 | −0.98 |
| | $T_{mrt}$ (°C) | 0.94 | 0.75 | 1.00 | −0.77 |
| **19:00** | PET (°C) | 1.00 | −0.06 | 0.99 | −0.34 |
| | V (m/s) | 0.62 | −0.21 | 0.60 | −0.14 |
| | $T_a$ (°C) | −0.06 | 1.00 | −0.19 | −0.15 |
| | $T_{mrt}$ (°C) | 0.99 | −0.19 | 1.00 | −0.32 |
| Strong correlation | | Moderate correlation | | Same parameters | |

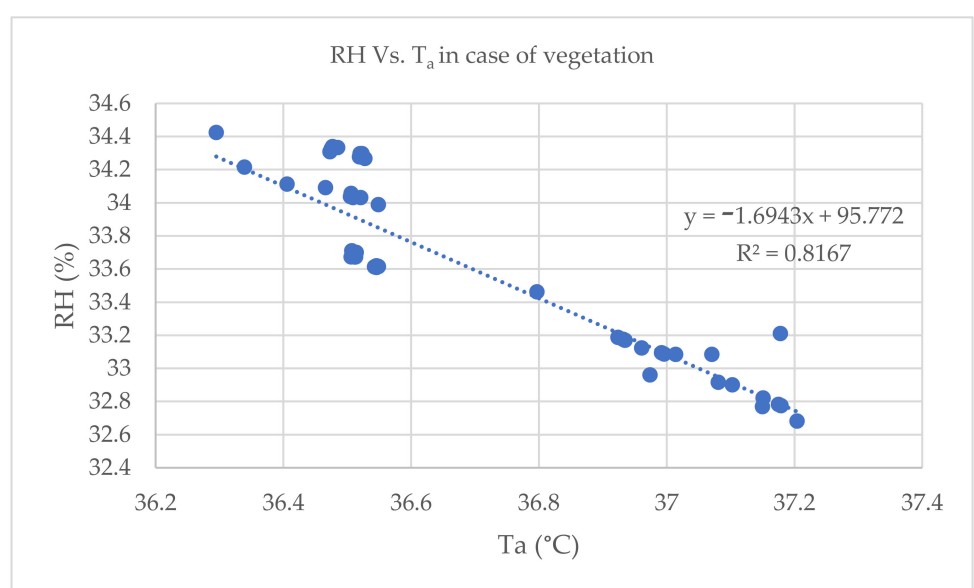

**Figure 10.** Correlation analysis at 11:00 in the case of vegetation, Relative humidity (RH) vs. Air temperature (Ta).

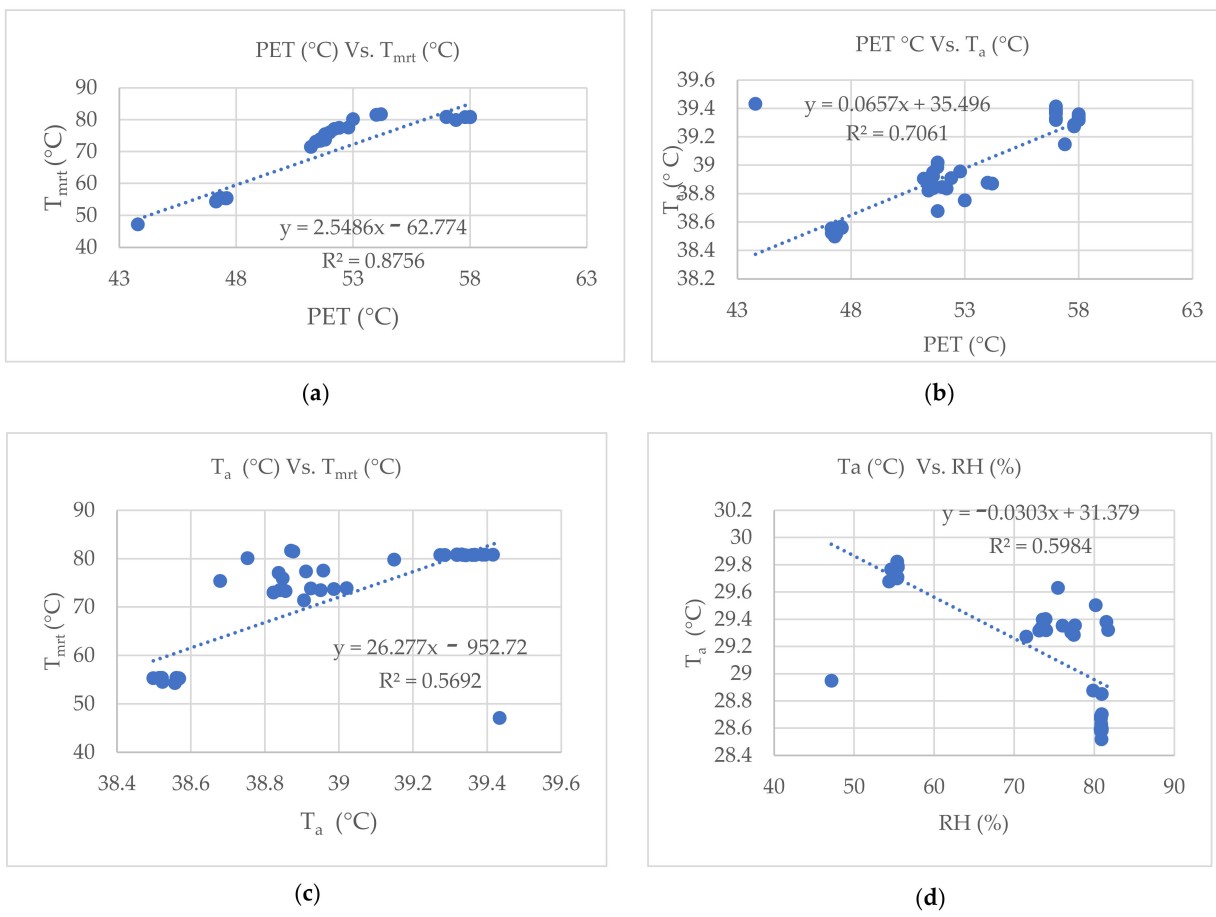

(**a**)                      (**b**)

(**c**)                      (**d**)

**Figure 11.** Correlational analysis at 16:00 in the case with vegetation: (**a**) PET vs. Tmrt; (**b**) PET vs. Ta; (**c**) Ta vs. Tmrt; and (**d**) Tmrt vs. RH.

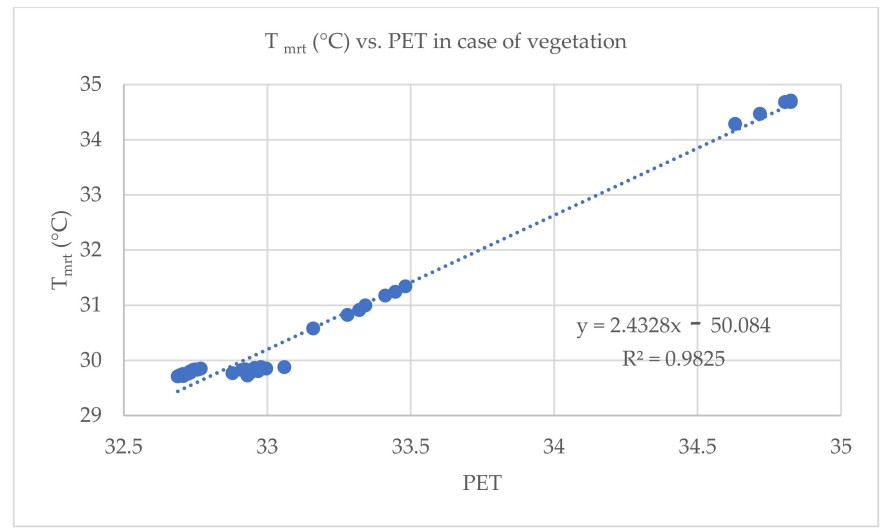

**Figure 12.** Correlation between Tmrt and PET parameters at 19:00 in the case with vegetation.

### 3.3.1. No Vegetation

A relationship between the human thermal comfort index (Physiological Equivalent Temperature, PET) and the mean radiant temperature at the times 11:00, 16:00, and 19:00 was observed; see Table 7. However, in the morning, the association was strong and inverse. No correlation was found between PET and RH at 11:00; however, an inverse correlation

was observed at 16:00 and 19:00 (−0.91 and −0.7, respectively). The relationship between PET and V fluctuated over the day, beginning with a strong inverse relationship at 11:00, no relation at 16:00, and a moderate relationship at night. The same pattern was seen between PET and $T_a$, where a tradeoff was clear at 16:00 and no relationship was observed during the day. Similarly, a tradeoff was clear between the air temperature and the mean radiant temperature only at 16:00. A different pattern of association was observed between $T_a$ and RH along the day, where strong negative relationships (−0.95, −1, and −0.95, respectively, for the three times) existed. Finally, the relationship between the wind speed and the mean radiant temperature varied over the day. A strong correlation was noted at 11:00, and then the degree of correlation decreased to moderate values (0.65 and 0.7 at 16:00 and 19:00, respectively).

### 3.3.2. With Vegetation

Study of the parameters Air temperature, Physiological equivalent temperature, and Mean radiant temperature (Table 8) indicated that they were intricately linked together at 16:00. Only at 19:00 was a strong relationship between the Radiant temperature and Physiological equivalent temperature maintained. At 16:00, there were tradeoffs between air temperature and PET (0.84), and between PET and RH (−0.89). One last change was observed between air temperature and relative humidity at 19:00, but no relationship was observed; however, strong negative correlations were observed at 11:00 and 16:00 (−0.9 and −0.97, respectively).

We took a closer look at the impact of vegetation on these parameters and analyzed the changes, from which the following was concluded:

At 11:00

- The major impact of vegetation was witnessed between $T_{mrt}$ with the PET index and V; the relationship between $T_{mrt}$ and PET changed from an inverse strong relation to a moderate relation. The inverse relationship was seen between $T_{mrt}$ and V, and the strong relationship was changed to no relationship when the vegetative pattern was present.
- The relationship between PET and V decreased to a moderate relationship (r = −0.65).
- A strong negative association developed between PET and $T_{mrt}$, to a positive moderate relationship (r = 0.54).
- Slight and negligible changes were witnessed between PET with RH and $T_a$.
- The lack of relationship between PET and V turned into a moderate correlation (0.51).
- Otherwise, relationships were maintained between the other parameters.

Generally, it is evident that the presence of vegetation contributed to changes in the relationships between parameters. The only reliable correlation that could be used further in the study was that between Air temperature and Relative humidity, in the case of vegetation and no vegetation (r = 0.9); see Figure 10:

$$y = -1.6943x + 95.772, \text{ where } y = Rh\% \text{ and } x = Ta$$

Furthermore, the PET and $T_{mrt}$ correlation could be used, in the case with vegetation:

$$y = 0.3035x + 30.034, \text{ where } y = T_{mrt} \text{ and } x = PET$$

At 16:00

- The strong associations between PET, $T_a$, $T_{mrt}$, and $T_a$ with $T_{mrt}$ were maintained.
- Moderate relationships between $T_{mrt}$ with V and RH were also maintained, when vegetation was considered.

Generally, it is evident that the presence of vegetation had a minor influence on the different parameters. Most of the relationships between the parameters were maintained. The following correlations are reliable and can be further used:

$$y = 2.5486x - 62.774, \text{ where } y = T_{mrt}, x = PET$$

$$y = 0.0657x + 35.496, \text{ where } y = Ta, x = PET$$

$$y = 26.277x - 952.72, \text{ where } y = Tmrt, x = Ta$$

$$y = -0.0303x + 31.379, \text{ where } y = Ta, x = RH$$

$$y = -0.0951x + 34.213, \text{ where } y = PET, x = RH$$

At 19:00

Major changes were witnessed when vegetation was present.

- It is notable that a strong correlation existed between PET and $T_{mrt}$ (0.99), as shown in Figure 11; differing from the case of no vegetation, where a moderate relation was observed between these parameters.
- Other parameters, such as PET and $T_a$, did not show any significant correlation (and PET and RH, in the no-vegetation case).
- There was no significant impact of vegetation on PET and V, and $T_{mrt}$ and V.
- Generally, there was a significant correlation between PET and the mean radiant temperature when vegetation was present, as shown in Figure 12.

$$y = 2.4328x - 50.084, \text{ where } y = T_{mrt}, x = PET$$

## 4. Discussion

For this research, we carried out a quantitative study to explore the solar irradiance reduction and overall cooling effect that vegetation provides in moderating microclimatic parameters—namely, Air Temperature ($T_a$), Mean Radiant Temperature ($T_{mrt}$), relative humidity (RH), and Physiological Equivalent Temperature (PET)—in urban spaces. The microclimate of urban spaces is highly dependent on: (a) The urban character, which includes the building geometry and the surface materials of pavements and buildings; and (b) the vegetative pattern, which considers multiple aspects in the case of trees (i.e., the tree geometry [12], spacing between trees, and foliage character [25,31]). The type of tree and its characteristics, such as density and spacing, have been discussed in regions with the same climatic zone, such as Dubai [25], and in different latitudes [16,24].

Egypt, especially the Greater Cairo Region, has witnessed notable urban vehicular network re-development, to facilitate mobility in the region. Changes in the street microclimate are expected to add to the UHI, as related to complicated urban issues (e.g., overpopulation, traffic, and urban emissions). Thus, landscapers, planners, and those in related fields have been urged to focus on the optimal green infrastructure that could be integrated into the design process, in order to enhance the overall climatic conditions of the city. This study focused on a mid-rise urban street in the eastern part of Cairo, located in the Sheraton Heliopolis area, representing a sample street (i.e., urban canyon) witnessing the re-development of the vehicular network.

For validation and calibration, as an initial step, we used the Meteonorm software for metrology data set calibration and to generate an EPW weather file. This method has been previously used in [37]. Then, the boundary condition, the urban geometry, existing and proposed planting patterns, and materials were also calibrated and set into a three-dimensional scaled model. Before simulating the developed plantation patterns in this study, on-site field (i.e., real-world) measurements were conducted, using metrological measurement devices to measure the air temperature, relative humidity, and wind speed.

ENVI-met simulation was performed on the same measurement day and time, that is, on 28th June 2021 from 12:00 to 22:00. The outcome of the simulation and the measurement

day was statistically compared, using correlational analysis involving the $R^2$, RMSE, and IA metrics. The results showed good agreement between measured parameters and ENVI-met simulation outputs, regarding air temperature and humidity, as with [32,37]; however, the insignificant calculation of the wind speed parameter was considered unreliable. This finding is similar to that of Elwy et al. [19], and could be attributed to the non-windy site conditions and the atypical extreme heatwave on the measuring day. Therefore, we used 24 h forced entry into the ENVI-met boundary conditions from the output exported from Meteonorm.

The current plantation design and no-vegetation scenarios were modeled in ENVI-met. Seven further scenarios were systematically designed, based on previous findings in the literature. Random and dispersed arrangements were discarded [24], as they had the smallest climate moderation, as previously mentioned in [36,39]. The minimum acceptable spacing between the trees was selected as 5 m, in order to ensure the optimal climate moderation (based on [24,25]), and with no overlap [36]. A small-sized tree was selected, to ensure the minimum temperature increase at night, based on [16], with a wide canopy to provide adequate shading. Thus, *Cassia nodosa* was used. Its characteristics were adopted from [29], and its LAD and LAI records were referred from [30], as appropriate measurements had been conducted.

Our initial finding was somewhat surprising, as the percentage of asphalt surfaces was relatively high, compared to the overall site area, and more than half the site had low and medium albedo surfaces. This, in turn, fostered low radiation absorption during daytime and gradual release at night [53]. Additionally, the overall air temperature increased, reaching 37, 38.8, and 35.14 °C at 11:00, 16:00, and 19:00, respectively. This directly elevated the PET ranges, reaching unprecedented levels. The presence of green infrastructure will enhance the microclimate; however, replacing dark-colored pavement with reflective or light-colored pavement will lead to better results. This finding has been discussed previously, in [30], where the surface and ambient temperatures could decrease by up to 7 °C.

Seven different tree patterns were proposed, with varying percentages (2%, 3%, and 4%). Grass was also added as a ground cover, elevating the percentages to 19%, 20%, and 21%. This green coverage is nearly half that indicated by [51], where a temperature reduction of 1 °C was described when the green coverage was 33% in high-density urban areas. However, observations were limited and could not be extrapolated to minor temperature increases when the scenario in which trees were clustered to the side and single trees were centered (max. 0.04 °C at 11:00, 0.11 °C at 16:00, and 0.06 °C at 19:00), despite the fact that this scenario had the highest vegetation percentage, compared to the other proposed scenarios.

We derived the spacing and arrangement of the trees, based on similar cases with the same climate character. Across the different proposed tree scenarios, a minor temperature moderation range was witnessed. This returns to the impact of the canopy size across the street, which was wide in this study. The wider the size of the canopy, the better the temperature reduction that could be observed, due to the triple effect of the canopy: Shading, evapotranspiration, and albedo increase [13]. The outcome of a small-scale outdoor experimentally developed model in China proved the same finding as that found in this study, as small-canopied street canyons, at the pedestrian level, elevate the temperature range, when compared to the use of large canopy sizes. Additionally, studies found that single-row big and tall trees can improve the environment [25,54]; this finding aligns with our finding in the base case/existing scenario, where big trees and tall palms exist within the considered wide street canyon. Sodoudi et al. have pointed out that the mean cooling response of small canopy trees is less than that of large canopy trees, across different urban configurations and temporal responses. However, one different finding has been witnessed in a 50 * 50 configuration at 5:00 a.m., where the small canopy performed better [55]. This is in alignment with the initial result of this study's findings. In wide streets, though, tall trees are preferred, due to their better temperature change effect in hot arid regions.

In terms of solar irradiance, by contrast, the arrangements recording the best mitigation (by 1% at 11:00) were single-side trees and clustered in the center, single centered trees, side staggered clustered trees, clustered trees in the center, single-side, and lastly single trees on the side after the vegetative case, which had 2% higher radiance reduction. Major variation was observed in the simulation outcomes at 16:00, between the proposed patterns and the existing vegetation pattern. A 10% reduction was witnessed with the current tree pattern, while 4% and 1% reductions were achieved with clustered trees on the side with single centered trees and single-side trees, respectively. This finding was expected, as the present trees and palms in the middle island have larger canopy width [23] and are taller [25] (11 and 9 m, respectively) evergreens with dense foliage (see the figures in Appendix B). This could be interpreted through the work of [12], where 80% of the direct radiation was absorbed and the rest was reflected, leading to a lower radiation range and a better comfort range.

By contrast, in terms of the behavior seen at 16:00, the mean radiant temperature varied considerably between when clustered trees were on the side with single trees in the center and the current arrangement, which has denser and larger foliage. Generally, at night, the trapped heat inside and under the canopy is released [22], while the arrangement [23], spacing between trees, height of the trees [25], presence of grass [16], and the wind flow [26,56] play important roles. In Scene 6, medium-sized trees are clustered on the side of the middle island and centered with 5 m spacing; this is considered a high vegetation percentage per m$^2$, which could limit and may block the wind flow, thus acting as a modifier. Although the trees are larger and have more foliage in the current design, sufficient space is present between the trees, allowing the wind to act effectively. This observation was proven with single-side trees and clustered trees in the center, where a lower vegetative percent was present.

Despite the differences between tree arrangements, locations, and sizes, the simulation result findings provide support that vegetation could extensively enhance human comfort in outdoor spaces, through shade provision and the evapotranspiration effect [30,32,36]. This was clearly witnessed at 11:00 and 16:00 in all scenarios; however, at night, the PET records were the best under the no-vegetation scenario.

The relative humidity correlated negatively with the air temperature in both no-vegetation and vegetation scenarios, as the percentage of the area shaded by the tree canopies and the buildings is lower than unshaded areas. However, under the tree canopy, regardless of the canopy size in all of the developed plantation patterns, the relative humidity was increased. This finding is in alignment with [32], due to the evapotranspiration effect the tree carries out under its canopy; in turn, the temperature range decreases. The moderate to strong correlation between $T_{MRT}$ and PET was expected under both vegetation and no-vegetation scenarios, as $T_{mrt}$ is a parameter that expresses the net radiation within the environment, as discussed in [57]. However, the negative correlation witnessed at 11:00 in the no-vegetation correlation matrix is still questionable.

Generally, there were changes in the wind pattern under the no-vegetation scenario. This finding can be attributed to the turbulence of the prevailing wind flow caused by the spacing present between existing buildings and the street intersecting with other streets. A strong significant correlation between the air temperature and the wind at 16:00, in contrast to the rest of the day, was because the wind speed was already low (i.e., at Receptor 2, in the no-vegetation scenario, the wind was 1.6 m/s, while the existing design elevated it to 1.8 m/s). Scenario 7, which has the highest tree number, decreased the wind to 1.4 m/s. Trees significantly modified the wind pattern: a major change was clearly seen in the existing case, as the trees in this case have wider canopies and are much taller than the proposed tree type in the developed design scenarios; this provides a better chance for the wind to flow and increase its speed, as previously stated by [56]. This finding also justifies the increase in temperature range within the different developed design scenarios, as wind transports the transpired air created under the tree canopy; however, static wind impedes this effect. The longwave radiation emitted from the surrounding surfaces and the tree

leaves add to the effect [16,21,30]. Finally, the weather conditions of the area, the percentage of vegetation, and the type of vegetation all have a great influence on the outcomes of the derived correlations between the parameters mentioned in Section 3.3.2.

## 5. Conclusions

This research constitutes an important contribution towards the integration of Green Infrastructure in the urban development field within the Greater Cairo Region for sustainable urban development. Although a notable vehicle re-development has been witnessed within the city, there is still a need for quantified and reliable guidelines for tree coverage, spacing, and arrangement, in order to ensure solar irradiance flux on the ground surface while having a positive implication for UHI mitigation. A mid-rise compact urban street was taken as a case study, as most of the redevelopment has been undertaken in similar areas.

Through the simulation of nine design patterns, we presented results that demonstrated that, in wide street canyons, the addition of trees generally produces an enhanced cooling capacity, contingent on: (1) The percentage of tree coverage, depending on the canopy size; (2) the spacing between trees; and (3) their arrangement [16,24,25,54,55]. However, when low albedo surfaces exceed 30%, reflective or light-colored pavements are recommended in the first place, as the surface and ambient temperature can be decreased [30] In this case, large spherical evergreen trees with a canopy coverage of 25 m [55] are recommended, considering that no overlap exists and the spacing between trees is sufficient to allow for air flow and the saturated vapor to flow around and between the trees. This may guarantee enhanced microclimate conditions at both day and night.

Evaluating the outcomes of the ENVI-met simulation for the existing case and the developed plantation patterns led to suggestions for a wide street canyon oriented in the northwest direction featuring mid-rise compact developments, as follows:

- In the western part of the street, where trees are placed next to the buildings' east side, the area is mostly shaded by the building throughout the day. The air temperature range is minorly changed by a maximum 0.016 °C and the mean radiant temperature by 28 K. Thus, grass would be recommended for climate moderation, or the present trees' size should be maintained, as larger canopies may hinder or trap the airflow.
- In the eastern part of the street, where trees are placed next to the buildings' west side, small-sized evergreen trees perform well throughout the day. The air temperature decreases by 0.1 °C and mean radiant temperature is lower by 26 K. An inverse outcome was observed in the lower part of the street, where the temperature increased by 0.14 °C. Thus, it is recommended that the upper evergreen is maintained, with the addition of grass surfaces to enhance the radiant temperature values. Meanwhile, in the lower part of the street, deciduous trees are suggested, with grass as a ground cover.
- In the middle island, among the different developed plantation patterns, the highest canopy coverage (4%) led to better mean radiant reduction: 1.3, 1, and 0.75 °C for the average receptor points under the single-side and cluster trees, cluster side trees with single center arrangement, and cluster side trees arrangements, respectively.
- Finally, to investigate the responses of trees under thermal extremes and heat waves under the same geometrical attributes, the correlational analysis mentioned earlier can be adopted for rapid prediction. This should be applied cautiously, as the outcome varies according to the weather conditions, tree percent, and its type. It is valid when considering high ambient temperatures, as the cooling capacity of trees is highly dependable on the saturation vapor pressure released during plant transpiration [58].

Further investigation should be undertaken, in order to explore the impact of large deciduous trees, which play distinguished roles in open recreational spaces and in wide streets with the same context conditions. It may be useful to study the optimized spacing between these trees and arrangement pattern (e.g., single- or double-row arrangements), in terms of where the shade flux reaches the asphalt surface and the parking lot.

We recommend that planners add a studied and planned Green Infrastructure network in alignment with the newly developed vehicular network. Simulation is highly recommended before on-site implementation.

**Author Contributions:** Conceptualization, M.F., W.M.E. and M.A.S.; methodology, M.F., W.M.E. and M.A.S.; software, W.M.E.; validation, W.M.E. and M.F.; formal analysis, W.M.E., M.A.S. and M.F.; investigation, W.M.E. and M.F.; resources, W.M.E., M.A.S., and M.F.; data curation, W.M.E., M.A.S. and M.F.; writing—original draft preparation, W.M.E., M.A.S. and M.F.; writing—review and editing, W.M.E., M.F. and M.A.S.; visualization, W.M.E., M.A.S. and M.F. All authors have read and agreed to the published version of the manuscript.

**Funding:** This research received no external funding.

**Institutional Review Board Statement:** Not applicable.

**Informed Consent Statement:** Not applicable.

**Data Availability Statement:** Not applicable.

**Conflicts of Interest:** The authors declare no conflict of interest.

**Abbreviations**

| | |
|---|---|
| MENA | Middle East and North Africa region |
| PET | Physiological Equivalent Temperature |
| $T_{mrt}$ | Mean Radiant Temperature |
| $T_a$ | Air Temperature |
| V | Wind speed |
| RH | Relative Humidity |
| TMY | Typical Meteorological Year |
| UHI | Urban Heat Island |
| TC | Tree Canopy |
| WMO | World Metrological Organization |
| EPW | Energy Plus Weather format |

**Appendix A**

**Table A1.** Summary of the temporal responses of temperature outputs for vegetation and no-vegetation scenarios.

| Time Frame | Scenario | Overall Min. $T_a$ | Overall Max. $T_a$. | Receptor 1 | Receptor 2 | Receptor 3 | Receptor 4 | Receptor 5 | Receptor 6 | Average |
|---|---|---|---|---|---|---|---|---|---|---|
| 11:00 | No-veg. | 36.11 | 38.11 | 36.54 | 36.46 | 36.93 | 36.50 | 36.40 | 37.08 | 36.65 |
| | Current design | 35.35 | 38.10 | 36.54 | 36.33 | 36.79 | 36.50 | 36.29 | 37.17 | 36.61 |
| | Δ $T_a$ | 0.76 | 0.01 | −0.003 | 0.127 | 0.135 | −0.001 | 0.111 | −0.097 | 0.045334 |
| 16:00 | No-veg. | 38.21 | 40.37 | 38.555 | 38.752 | 39.261 | 38.507 | 38.717 | 39.294 | 38.84 |
| | Current design | 38.14 | 40.29 | 38.556 | 38.752 | 39.148 | 38.523 | 38.678 | 39.433 | 38.84 |
| | Δ $T_a$ | 0.07 | 0.08 | −0.001 | 0 | 0.113 | −0.016 | 0.039 | −0.139 | 0 |
| 19:00 | No-veg. | 34.60 | 35.40 | 35.082 | 35.051 | 35.284 | 35.07 | 35.072 | 35.288 | 35.14 |
| | Current design | 34.67 | 35.40 | 35.073 | 35.064 | 35.264 | 35.075 | 35.085 | 35.258 | 35.136 |
| | Δ $T_a$ | −0.07 | 0 | 0.009 | −0.013 | 0.02 | −0.005 | −0.013 | 0.03 | 0.0046 |

**Table A2.** Metrological data for the scenarios.

| Time Frame | Scenario | Overall Min. Temperature | Overall, Max. Temp. | Receptor 1 | Receptor 2 | Receptor 3 | Receptor 4 | Receptor 5 | Receptor 6 | Average |
|---|---|---|---|---|---|---|---|---|---|---|
| 11:00 | Sc1 | 36.10 | 38.12 | 36.548 | 36.519 | 36.961 | 36.512 | 36.505 | 37.151 | 36.69933 |
| | Sc2 | 36.10 | 38.12 | 36.546 | 36.475 | 36.924 | 36.51 | 36.477 | 37.103 | 36.6725 |
| | Sc3 | 36.09 | 38.12 | 36.549 | 36.523 | 36.996 | 36.513 | 36.509 | 37.179 | 36.7115 |
| | Sc4 | 36.10 | 38.12 | 36.546 | 36.485 | 36.935 | 36.51 | 36.473 | 37.071 | 36.67 |
| | Sc5 | 36.10 | 38.12 | 36.547 | 36.52 | 36.974 | 36.511 | 36.507 | 37.15 | 36.7015 |
| | Sc6 | 36.10 | 38.12 | 36.549 | 36.549 | 37.014 | 36.514 | 36.528 | 37.204 | 36.72633 |
| | Sc7 | 36.10 | 38.12 | 36.548 | 36.521 | 36.992 | 36.512 | 36.506 | 37.175 | 36.709 |
| | No-veg. | 36.11 | 38.11 | 36.54 | 36.46 | 36.93 | 36.50 | 36.40 | 37.08 | 36.65 |
| 16:00 | Sc1 | 38.20 | 40.37 | 38.565 | 38.91 | 39.319 | 38.518 | 38.87 | 39.367 | 38.92483 |
| | Sc2 | 38.21 | 40.37 | 38.56 | 38.822 | 39.273 | 38.514 | 38.847 | 39.318 | 38.889 |
| | Sc3 | 38.20 | 40.37 | 38.566 | 38.986 | 39.342 | 38.521 | 38.839 | 39.394 | 38.94133 |
| | Sc4 | 38.20 | 40.37 | 38.561 | 38.855 | 39.285 | 38.515 | 38.836 | 39.331 | 38.89717 |
| | Sc5 | 38.20 | 40.37 | 38.564 | 38.95 | 39.318 | 38.518 | 38.876 | 39.366 | 38.932 |
| | Sc6 | 38.20 | 40.37 | 38.568 | 39.02 | 39.359 | 38.498 | 38.905 | 39.415 | 38.96083 |
| | Sc7 | 38.20 | 40.37 | 38.566 | 38.957 | 39.334 | 38.52 | 38.923 | 39.386 | 38.94767 |
| | No-veg. | 38.20 | 40.37 | 38.555 | 38.752 | 39.261 | 38.507 | 38.717 | 39.294 | 38.84767 |
| 19:00 | Sc1 | 34.69 | 35.43 | 32.688 | 33.411 | 32.956 | 32.707 | 33.447 | 32.931 | 33.023 |
| | Sc2 | 34.69 | 35.42 | 32.741 | 34.718 | 32.919 | 32.762 | 33.321 | 32.996 | 33.24 |
| | Sc3 | 34.70 | 35.44 | 32.741 | 34.718 | 32.919 | 32.762 | 33.321 | 32.996 | 33.24 |
| | Sc4 | 34.69 | 35.42 | 32.696 | 33.16 | 32.879 | 32.723 | 33.28 | 32.94 | 32.946 |
| | Sc5 | 34.69 | 35.43 | 32.706 | 34.631 | 32.912 | 32.732 | 33.342 | 32.968 | 33.215 |
| | Sc6 | 34.70 | 35.45 | 32.748 | 34.825 | 32.978 | 32.768 | 34.825 | 33.06 | 33.534 |
| | Sc7 | 34.70 | 35.44 | 32.731 | 33.482 | 32.924 | 32.756 | 34.805 | 32.972 | 33.278 |
| | | | | Mean Radiant Temperature | | | | | | |
| 11:00 | Sc1 | 71.94 | 77.1 | 74.029 | 76.031 | 73.797 | 74.074 | 75.975 | 75.732 | 74.93967 |
| | Sc2 | 71.99 | 77.04 | 74.132 | 72.116 | 73.899 | 74.176 | 75.715 | 75.65 | 74.28133 |
| | Sc3 | 71.88 | 77.21 | 74.056 | 75.74 | 73.772 | 74.127 | 75.684 | 75.668 | 74.84117 |
| | Sc4 | 71.98 | 77 | 74.102 | 75.71 | 73.859 | 74.171 | 75.65 | 74.216 | 74.618 |
| | Sc5 | 71.94 | 77.16 | 74.061 | 75.827 | 73.835 | 74.132 | 75.769 | 75.701 | 74.8875 |
| | Sc6 | 71.84 | 77.21 | 74.012 | 76.703 | 73.761 | 74.082 | 72.942 | 75.622 | 74.52033 |
| | Sc7 | 71.89 | 77.38 | 74.045 | 72.974 | 73.816 | 74.113 | 72.913 | 75.665 | 73.921 |
| 16:00 | Sc1 | 55.01 | 82.08 | 55.294 | 77.439 | 80.855 | 55.335 | 81.701 | 80.838 | 71.91033 |
| | Sc2 | 55.06 | 81.86 | 55.391 | 73.088 | 80.804 | 55.429 | 76.008 | 80.912 | 70.272 |
| | Sc3 | 54.99 | 82.19 | 55.341 | 73.769 | 80.785 | 55.417 | 73.546 | 80.904 | 69.96033 |
| | Sc4 | 55.05 | 81.84 | 55.368 | 73.341 | 80.825 | 55.44 | 77.102 | 80.92 | 70.49933 |
| | Sc5 | 55.02 | 82.13 | 55.326 | 73.567 | 80.825 | 55.401 | 81.503 | 80.903 | 71.25417 |
| | Sc6 | 54.97 | 82.21 | 55.319 | 73.965 | 80.799 | 55.369 | 71.478 | 80.885 | 69.63583 |
| | Sc7 | 55.01 | 82.1 | 55.347 | 77.572 | 80.841 | 55.419 | 73.896 | 80.905 | 70.66333 |
| 19:00 | Sc1 | 29.51 | 34.10 | 29.709 | 31.177 | 29.862 | 29.718 | 31.241 | 29.727 | 30.239 |
| | Sc2 | 29.49 | 33.77 | 29.825 | 34.472 | 29.823 | 29.843 | 30.915 | 29.851 | 30.78817 |
| | Sc3 | 29.53 | 34.81 | 29.825 | 34.472 | 29.823 | 29.843 | 30.915 | 29.851 | 30.78817 |
| | Sc4 | 29.49 | 34.09 | 29.732 | 30.581 | 29.767 | 29.76 | 30.826 | 29.772 | 30.073 |
| | Sc5 | 29.51 | 34.30 | 29.751 | 34.286 | 29.825 | 29.777 | 30.995 | 29.806 | 30.74 |
| | Sc6 | 29.54 | 34.73 | 29.836 | 34.684 | 29.877 | 29.854 | 34.709 | 29.881 | 31.4735 |
| | Sc7 | 29.53 | 34.70 | 29.801 | 31.344 | 29.835 | 29.829 | 34.684 | 29.835 | 30.888 |

## Appendix B

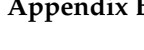

**Figure A1.** T$_{mrt}$ at 11:00 a.m. Cut level is 1.5 m above the ground. k = 2. The position of the view plan in Leonardo is 2 units above the ground.

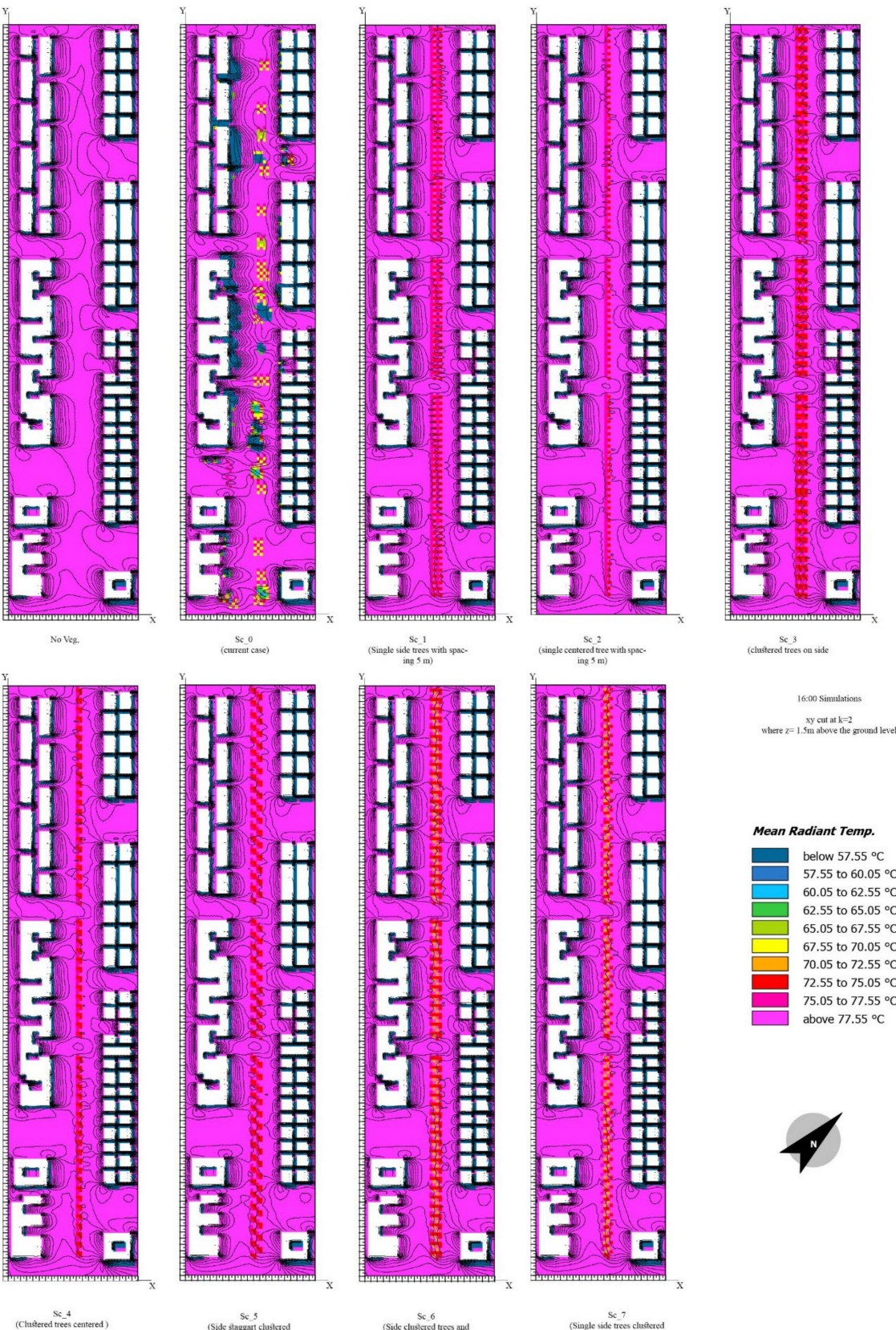

**Figure A2.** T$_{mrt}$ for 16:00. Cut level is 1.5 m above the ground. k = 2. The position of the view plan in Leonardo is 2 units above the ground.

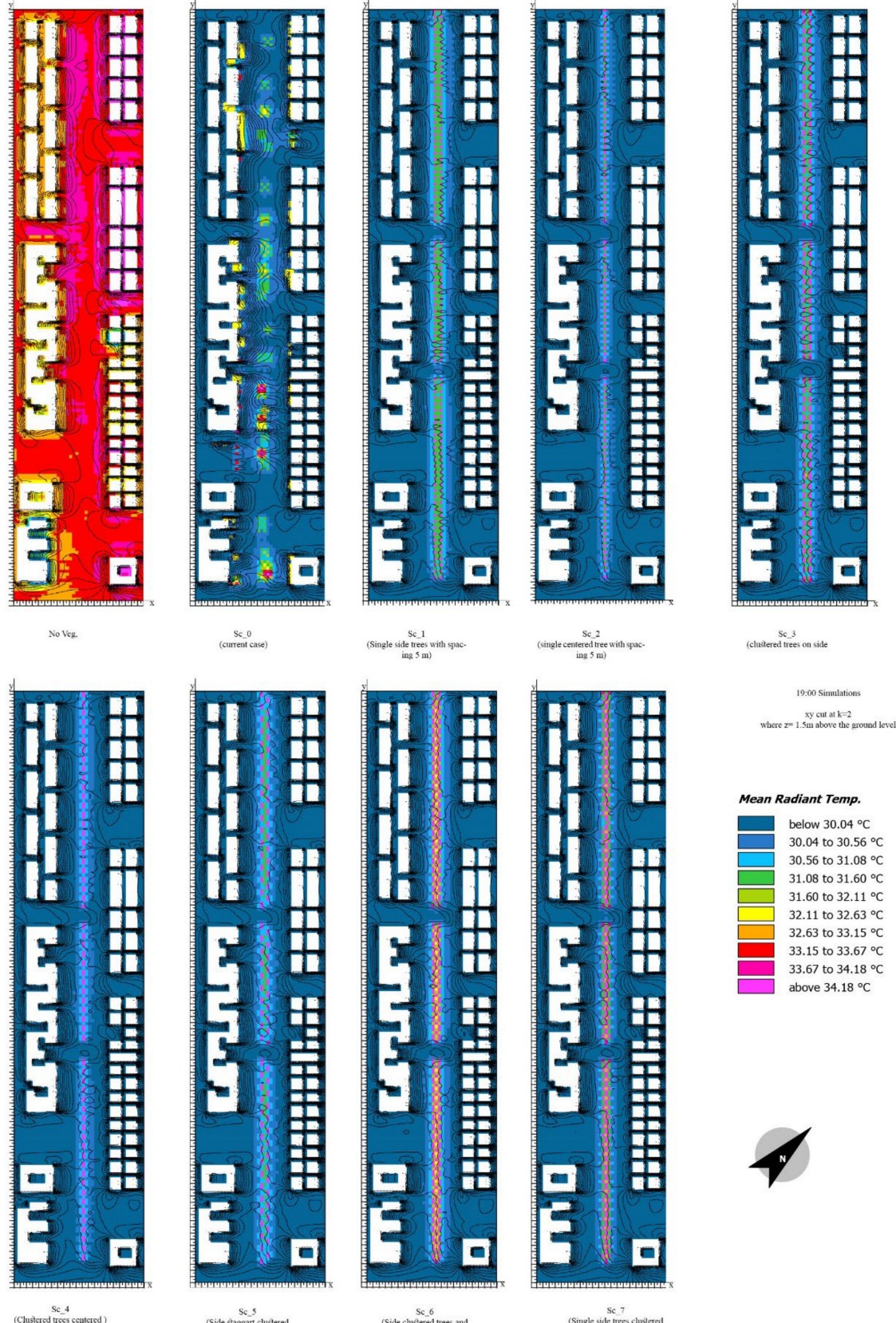

**Figure A3.** T$_{mrt}$ for 19:00. Cut level is 1.5 m above the ground. k = 2. The position of the view plan in Leonardo is 2 units above the ground.

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
