# Peer review of "Solar Irradiance Reduction Using Optimized Green Infrastructure in Arid Hot Regions: A Case Study in El-Nozha District, Cairo, Egypt"

_sustainability, doi:10.3390/su13179617_

Round 1
Reviewer 1 Report
Please see the attached.

Author Response
The authors are grateful for the positive and constructive comments of all reviewers which indeed increased the overall quality of the manuscript. We respect all comments and we tried to respond to each point carefully, however, very few comments have been explained in the following table without modifications in the manuscript itself.
Please see the attachment

Reviewer 2 Report
The title is informative and relevant together with the definition of the main aims and methods of the case study analyses and the simulation model worked out for green infrastructure improvement. References offer a good overview of relevant research papers in urban climate and green infrastructure aspects. Many GI papers underline the UHI mitigation capacity of urban green in the Western world where the UGI concept launched a hot topic in sustainability and urban landscape planning there is not enough knowledge about the hot climate regions in this aspect.
The research question and the method chosen for answering are clear. The authors developed a reasonable model for analyses of urban climate and urban green connections. The chosen site offers a reliable case study development. ENVI-met simulation software helped the case study analyses. The collected data and the evaluation are relevant to arrive at valuable results.
A detailed site map – built-in system, open space system, details of tree stock, surface material, and albedos – would help a lot to understand the urban context of the case study site. Only one site photo is given.
Results introduction is well structured and detailed.
The high proportion of asphalt surfaces – around 2/3 with low and medium albedo – causes a low radiation absorption in the daytime and high back radiation in the nighttime, as it is reflected in the PET ranges. This result should call attention to a further comparative site analysis where the contra-site may have better albedo and radiation – back-radiation grades.
The conclusion part would need one more aspect to highlight or to propose further research, namely the development ideas and issues based on this research for the development of urban planning tools and methods, for implementing urban green infrastructure and climate change mitigation strategy.
Major points in the article need clarification, refinement, and/or additional information and suggestions for what could be done to improve the article.
- some missing aspects and definitions and measurements in Tree character box: Vertical size (m); Crown density (%?); health and condition aspects are missing though tree canopy density is in a close connection with the overall health and condition of the tree; In Urban character box: surface material is missing; in Climate character box: albedo is missing
- Based on European urban climate researches the albedo given in Table4 seems questionable. The European literature offers the albedo of asphalt surfaces as 0,05 - 0,2 and for grass vegetation 0,2-0,4 depending on the quality, density, and the watering system. Hardly believable that yellow tiles are 2,5 better in the albedo context than a green, living, and though on a low scale, still transpiration surface. Please, check these data.
- The conclusion part should offer more dynamic proposals both for further researches in this context and for the introduction of urban green infrastructure strategy.
Minor points: a missing reference, line225
Not having native English knowledge I am not fully capable to decide about the quality of translation. Still, I found some hard-to-read paragraphs or sentences in the text and some debatable expressions, professional terms. Without giving the full list of problematic parts, I call the authors’ attention to a thorough English language check to help readers in clear understanding.
As for the terms, the definitions and phrases should be checked too.
Some examples:
Climate nature (line51) is more climate system or simple climate either local or regional level.
Green infra (from the title on) would be better to use the correct form, green infrastructure GI as it was stated and formed in the European strategy paper and since then, in all research papers in this context.
Shade trees (line 65 and more) is better to call shading trees or shade-providing trees.
Tree grouping (line106) might be better a planting system or tree plantation type or tree planting pattern.
Latin plant names writing is regulated: Cassia nodosa (capital only in the first taxon name). Cassia nodosa is native in Southeast Asia, in the tropical region. According to your home professional literature, this tree seems a common species, though not native. I have seen the different ways of Latin name writing so please free to decide which form to use.
Author Response

(The authors gave the same response as above.)

Reviewer 3 Report
The manuscript entitled “Solar Irradiance Reduction Using Optimized Green Infra in Arid- Hot Regions: A Case Study in El-Nozha District, Cairo, Egypt” by Elbardisy et al. studied the effect of urban trees in regulating urban heat stress using numerical simulations. The authors designed a set of numerical scenarios and analyzed the effectiveness of urban trees through comparisons. One of my major concerns is the details of the model. The authors will need to detail how the urban trees are numerically represented in the model and if this is sufficient to realistically represent the physical processes such as ET and shading. On the other hand, the authors did not document the sensitivity of their simulations. Did the authors conduct any pretest to make sure the results are stable, and the spatial resolution is sufficient? Here are some additional comments:
Line 10: Please spell out “MENA”.
Lines 64-65: Trees “modulate solar radiation released through shade and evapotranspiration”? Solar radiation is not released through shade or evapotranspiration.
Lines 108-110: “However, the tapped heat slowly releases in nighttime promoting higher humidity levels in the context [20].” – the trapping effect of tree canopies can also lead to higher surface and air temperatures beneath trees (see e.g., https://doi.org/10.1016/j.buildenv.2011.06.025).
Lines 145-149: Please consider removing the entire paragraph as the focus of this study is on cooling instead of pollution dispersion.
Lines 151-167: the authors will need to review existing studies in this city and knowledge gaps, and explain why this study is needed (e.g., why El-Nozha district? What is unknown?).
Lines 272-273: Already mentioned above.
Section 2.2.3: The authors should document the location and height of measurements and receptors in ENVI-met.
Figure 7: The authors will need to justify the selection of these six receptor locations (rather than other locations).
Lines 441-442: “Lastly, noted relationship between wind speed and the radiant temperature at 11 and 19:00.” Statements like this one are rather unclear – what should the readers note?
Figures 9-10: axis titles are missing.
Section 4: the authors will need to summarize and explain the observed statistical relationships in Section 3.3. Currently the authors only described the results.
Lines 592-612: All these “rapid prediction” equations are conditioned on the selected day in summer, while on other days these relationships will likely change.
Section 5: The authors will need to mention the limitations of this study. For example, the cooling effect of urban trees also varies with ambient temperature as observed in e.g., https://doi.org/10.1016/j.rse.2019.03.024.
Author Response
The authors are grateful for the positive and constructive comments of all reviewers which indeed increased the overall quality of the manuscript. We respect all comments and we tried to respond to each point carefully, however, some very few comments have been explained in the following table without modifications in the manuscript itself.
Please see the attachment

Round 2
Reviewer 1 Report
Please see the attached.

Author Response
The authors are grateful for the positive and constructive comments of all reviewers which indeed increased the overall quality of the manuscript.
Please see the attachment

Reviewer 3 Report
The manuscript has been substantially improved in this round of revision. Thanks for the effort. Following my previous comments, I suggest the authors make one additional change: Please point out that the prediction equations in Section 3.3.2 can vary under different weather conditions and should be applied with cautions in Discussion and Conclusions.
Author Response
The authors are grateful for the positive and constructive comments of all reviewers which indeed increased the overall quality of the manuscript.
Please see the attachement
